



# Characterising Marine Heatwaves in the Svalbard Archipelago and Surrounding Seas

Marianne Williams-Kerslake[1,2], Helene R Langehaug[1,2], Ragnheid Skogseth[3], Frank Nilsen[2,3], Annette Samuelsen[1], Silvana Gonzalez[4], and Noel Keenlyside[2]

[1]Nansen Environmental and Remote Sensing Centre (NERSC), Bjerknes Centre for Climate Research, Jahnebakken 3, 5007, Bergen, Norway
[2]University of Bergen Geophysical Institute, Jahnebakken 3, 5007, Bergen, Norway
[3]University Center in Svalbard (UNIS), Longyearbyen, Svalbard, Norway
[4]Institute of Marine Research (IMR), Nordnesgaten 50, 5005, Bergen, Norway

**Correspondence:** Marianne Williams-Kerslake (marianne.williams-kerslake@nersc.no)

**Abstract.** In the Arctic Ocean, satellite-based sea surface temperature data shows that marine heatwave (MHW) intensity, frequency, duration, and coverage have increased significantly in recent decades, raising concern for Arctic ecosystems. A high frequency (more than three events per year) of MHWs has been shown around the Svalbard Archipelago. Based on this, we investigate MHW trends around Svalbard at the surface and subsurface, using a regional reanalysis from TOPAZ (1991—2022). We find an increase in the frequency and duration of MHW events around the Svalbard Archipelago over the last decade. Furthermore, we observe an increase in MHW frequency and duration west of Svalbard, associated with a long-term rise in sea surface temperature in the region. Analysis of eight individual summer (June-September) MHW events lasting longer than 10 days in Svalbard West, indicated the presence of four shallow ($\leq 50\,\mathrm{m}$) and four deep ($> 50\,\mathrm{m}$) MHWs after 2010, with a mean duration of 29 days. Some events extended from Svalbard West into the Barents Sea. Heat budget analysis demonstrated a greater contribution of ocean heat transport compared to air-sea heat fluxes in driving the MHW events. Deep and shallow events were associated with ocean heat transport anomalies of up to 9TW. This new understanding of MHW characteristics including their horizontal and vertical distribution is key to assessing ecological impacts.

## 1 Introduction

Marine heatwaves (MHWs) are characterised as prolonged periods of extreme high sea surface temperatures relative to the long-term mean daily seasonal cycle. MHWs have become more frequent due to climate change and this trend is likely to increase in the future (IPCC, 2021). Recent studies have noted that the average global annual MHW frequency and duration have increased by 34 and 17%, respectively over the last century (Oliver et al., 2018). Severe MHWs have been detected worldwide including in the Mediterranean Sea (Ibrahim et al., 2021), northwest Atlantic Ocean (Chen et al., 2014), China Seas (Li et al., 2019), and over the Red Sea (Mohamed et al., 2021). MHWs can be triggered by both local and remote atmospheric and oceanic forcings, including heat advection by ocean currents (Oliver et al., 2018), or atmospheric overheating through an anomalous air-sea heat flux (Olita et al., 2007; Chen et al., 2015).



An increase in the number and intensity of MHWs is a growing concern due to the threat to marine communities. The biological impacts of MHWs include; changes in species distribution, loss of biodiversity, and a collapse of habitat-forming

foundation species, including reef-building corals, seagrasses, and seaweeds (Smith et al., 2023). Recent MHW events have already been connected to mass mortality events of primary producers, corals and invertebrates (Filbee-Dexter et al., 2020; Garrabou et al., 2022; Shanks et al., 2020). Furthermore, studies predict that MHWs will cause a decrease in the biomass of commercially important fish species (Cheung et al., 2021). The latter are expected to have drastic socioeconomic impacts as the frequency of MHWs increases; the extent to which the impacts of MHWs are felt by fish species, however, remains unclear

(Fredston et al., 2023). Additionally, MHWs have been shown to impact weather patterns, with evidence showing an intensification of storms during MHW events at lower latitudes (Choi et al., 2024).

Studies have shown increasing trends in the annual intensity, frequency, duration, and areal coverage of MHWs in the Arctic Ocean (north of 60°N, Huang et al. (2021b)). The increase in MHWs in the Arctic Ocean has been linked to a rise in surface

air temperatures and sea ice retreat (Huang et al., 2021b). As a result of the phenomenon known as Arctic amplification, near-surface air temperatures in the Arctic have warmed faster by a factor of three to four, compared to the global average (Rantanen et al., 2022). Meanwhile, the annual mean sea-ice extent in the Arctic Ocean has decreased by 20% since the 1980s, with the largest decline observed in summer (Eisenman et al., 2011; Stroeve et al., 2025); and Arctic sea ice continues to decrease both in extent and thickness (Sumata et al., 2023). Unlike regions further south, knowledge of MHWs in the Arctic Ocean is limited

and we lack a full understanding of the triggers of Arctic MHWs and their impacts on biogeochemistry (report by PlanMiljø (2022) for the Norwegian Environment Agency).

The Svalbard Archipelago located north of the Arctic Circle at 74°–81°N, 10°–35°E (Fig. 1), has been shown to experience a relatively high frequency of MHWs compared to other regions of the Arctic (approximately 2-3 events per year, Huang

et al. (2021b)). The Svalbard Archipelago forms part of the Barents Sea shelf with an average depth of 200-300m. Adjacent off-shelf areas exceed depths of 2000m. The region is influenced by cold Arctic water masses formed locally in winter and advected from the northeast by the East Spitsbergen Current and further by the Spistbergen Polar Current along the coast west of Svalbard, and the warm West Spitsbergen Current (WSC) which flows northwards transporting warm Atlantic Water (AW) along the slope of the West Spitsbergen Shelf (Fig. 1). The WSC continues along the West Spitsbergen shelf-slope and then

flows into the Arctic Ocean, where it circulates cyclonically.

Warm AW transported by the WSC is important for shaping the climatic conditions of the Svalbard Archipelago. Warming AW and a rise in regional sea temperatures, intensified by the loss of sea ice, are potential reasons for the high frequency of MHWs around Svalbard. AW transported by the WSC past Svalbard has warmed, with a positive trend of 0.06°C year$^{-1}$

from 1997-2010 (Beszczynska-Möller et al., 2012), with a particular increase observed from 2004-2006 (Walczowski, 2014). Furthermore, the presence of AW on the West Spitsbergen Shelf has increased during winter (Cottier et al., 2007; Nilsen et al.,



2016) and summer where an $8\% \, y^{-1}$ increase in the volume fraction of AW on the shelf southwest of Spitsbergen has been observed (Strzelewicz et al., 2022). Additionally, a shoaling of AW has been observed on the shelf and into the West Spitsbergen fjords including Kongsfjorden (Tverberg et al., 2019) and Isfjorden (Skogseth et al., 2020). Meanwhile, the maximum temperatures in Isfjorden on the west coast of Svalbard have increased by about 2°C during the last hundred years (1912-2019) (Bloshkina et al. (2021), for location see Fig. 1). The largest increase has been observed in Isfjorden with an increase in summer and winter SST of 0.7 ± 0.1 °C per decade since 1987 (Skogseth et al., 2020). Furthermore, in Isfjorden, positive trends in volume weighted temperature and volume weighted salinity, suggesting more AW inflow, are found both in summer and winter (Skogseth et al., 2020). Understanding the drivers and characteristics of MHWs around Svalbard is essential as commercial fishing is carried out annually from the southern border of the Svalbard zone at 74°N, and around the Svalbard Archipelago up to about 81°30' N (Misund et al., 2016), making it a region of high economic importance.

This study identifies MHW events in 1991-2022, around the Svalbard Archipelago, by using surface and subsurface ocean temperature data from a physical reanalysis of the North Atlantic and Arctic region - TOPAZ4b (Xie et al., 2017). By comparing TOPAZ with observational data, the study evaluates how accurately TOPAZ captures MHW events around the Svalbard Archipelago. The primary objective of this study is to determine the characteristics of each MHW event, including its duration, intensity and spatial extent. Furthermore, the study explores the environmental factors contributing to the high frequency of MHWs in the Svalbard region.







**Figure 1.** Schematic map of the oceanic circulation around the Svalbard Archipelago; NwASC (Norwegian Atlantic Slope Current), NwAFC (Norwegian Atlantic Front Current), WSC (West Spitsbergen Current), SPC (Spitsbergen Polar Current), ESC (East Spitsbergen Current). Blue arrows represent cold Arctic water masses, red arrows represent warm Atlantic water masses (adapted from Vihtakari et al. (2019) and Eriksen et al. (2018)). Location of Rijpfjorden (R), Kongsfjorden (K), Isfjorden (IS) and Svalbard West domain (black box, 77-80°N, 5-15°E) are shown. Locations in the pathway of the NwAFC (green dashed box) and NwASC (Barents Sea Opening, pink dashed box) used for the analysis of MHW drivers are also shown. Small inset map indicates the locations of the Isfjorden mouth mooring (ISM), TOPAZ point for validation with the ISM mooring (TP1: 78.125°N,11.75°E), Yermak Plateau Mooring (YPM), Storfjorden Moorings (M1,M2) and M4 mooring. The TOPAZ bathymetry is shown.





## 2 Methods

Marine heatwaves (MHWs) are detected in the Svalbard Archipelago and surrounding seas using a physical reanalysis from TOPAZ. TOPAZ has been previously evaluated for the Arctic Ocean against a suite of ocean observations (Lien et al. (2016), Xie et al. (2019), Xie et al. (2023)). However, since this study focuses on a smaller region and looks at more local events in the Svalbard Archipelago, we have chosen to evaluate how accurately TOPAZ represents temperature on the shelves and in the fjords of Svalbard. The reanalysis is compared with oceanographic mooring data from the Svalbard Archipelago. The

observational datasets used are not assimilated to produce the reanalysis and thus are well suited to evaluate TOPAZ. Through comparison with mooring data, TOPAZ is shown to perform best at depth west of Svalbard, hence we investigate temperature trends and MHW events in this region in more detail. For this study, this region is termed Svalbard West (77-80°N, 5-15°E). Surface patterns in MHWs in TOPAZ are validated using satellite data.

### 2.1 MHW Definition and Metrics

Adapting methods from Hobday et al. (2016), MHWs have been detected when the TOPAZ daily sea surface temperature (SST) exceeds the $90^{th}$ percentile for at least 5 consecutive days, allowing no more than two days below the threshold within the 5 days. Successive events separated by gaps of two days or fewer were considered part of the same MHW. MHW events and their characteristics were determined using the Python marineHeatWaves module (*https://github.com/ecjoliver/marineHeatWaves/tree/master*). According to Hobday et al. (2016), the baseline SST climatology for the percentile should be based on at least 30 years of data.

For this study, the $90^{th}$ percentile threshold has been calculated for each grid cell of each calendar day of the year using daily temperature data over 32 years (1991–2022) as a fixed climatological baseline. For each calendar day, temperatures from a 5-day window centred on that day were used to determine the percentile. Hobday et al. (2016) suggests that daily, threshold time series may need to be smoothed to extract a useful climatology from inherently variable data. Consequently, we applied a 31-day moving window to smooth the $90^{th}$ percentile. MHWs were detected for the Svalbard West region by averaging SSTs

over the region and identifying periods when this regional mean exceeded the $90^{th}$ percentile. For the analysis of individual MHW events, we selected prolonged MHW events lasting at least 10 days.

In this study we have also detected subsurface MHWs, by determining to which depths temperatures exceed the $90^{th}$ percentile during each detected surface MHW event. Events shallower or equal to 50m were defined as shallow events. Those that

extended from the surface to deeper than 50m were defined as deep events.

The mean start date for MHWs in the Arctic Ocean (1982-2020) has been reported to be in August (Huang et al., 2021b); therefore, we focused on MHWs initiated during the Arctic summer period from June to September.

As in Huang et al. (2021b), each MHW event is described by a set of metrics (Hobday et al., 2016, 2018): Mean intensity (°C) is the average SST anomaly (SSTA) over the duration of the event. Maximum intensity (°C) is the highest SSTA during an





event. The event peak indicates the date of peak intensity. The duration (in days) of a MHW is calculated as the time interval between the start and end times and frequency (in events) is the number of events that occurred in each year.

Each MHW event is assigned a category defined by Hobday et al. (2018) based on its intensity. Each category is determined by the degree to which temperatures exceed the local climatology by looking at multiples of the 90[th] percentile difference (2× twice, 3× three times, etc.) from the mean climatology. The categories are as follows: moderate (Category I), strong (Category II), severe (Category III), and extreme (Category IV). Assigning categories to MHW events can be extremely useful as it enables comparison of events across different regions (Hobday et al., 2018). Furthermore, assigning events with a unique

identifier facilitates communication among experts and the general public (Nairn and Fawcett, 2013).

## 2.2  Ocean Heat Budget

Using methods adapted from Bianco et al. (2024) the ocean heat budget was determined for Svalbard West (77-80°N, 5-15°E). Considering a control ocean volume with surface A and vertical section S, where mass and salinity are conserved, the heat budget is given by the balance between advective and vertical heat flux terms. As in Bianco et al. (2024), we omit the lateral

heat diffusion term as this is negligible compared to the surface and advective flux terms (Lique and Steele, 2013):

$$
\underbrace{\frac{\partial Q}{\partial t}}_{Q_t} = \underbrace{\rho_0 C_p \int_S V T \, dS}_{\text{OHT}} + \underbrace{\int_A Q_s \, dA}_{\text{SHF}}, \tag{1}
$$

$Q_t$ is the ocean heat content tendency; OHT represents the advective ocean heat transport through S; $c_p$ is the specific heat capacity of seawater (3987 $J/kg \cdot {}^\circ C$) and $\rho$ is the density of seawater (1000 $kg/m^3$); V and T represent the cross-sectional velocity and potential temperature, respectively; SHF indicates the net sea surface heat flux, $Q_s$, over the surface A.


The net ocean heat transport into the Svalbard West domain is computed along each boundary - facing north, south, west, (the eastern boundary is excluded from the ocean heat transport calculation as it is bounded by land) at daily frequency,

$$
\text{OHT} = \rho_0 C_p \int_{-z(\lambda)}^{\eta} \int_{\lambda_1}^{\lambda_2} V(T - T_{\text{ref}}) \, dz \, d\lambda, \tag{2}
$$

where $\lambda_1$ and $\lambda_2$ are the coordinates of the section line, and $z(\lambda)$ is the depth at each section. A reference temperature

of -0.1°C was used. We consider the net SHF term over the Svalbard West box area. SHF is represented by the *surface downward heat flux in seawater* product in TOPAZ and consists of combined solar irradiance, sensible heat flux, latent heat flux, and longwave radiation. The OHT was calculated using this code - *https://github.com/nansencenter/NERSC-HYCOM-CICE/tree/master/hycom/MSCPROGS/src/Section.*



### 2.3 Ocean Heat Content

We computed the daily ocean heat content (OHC) of the upper 300 m using methods adapted from McAdam et al. (2023),

$$c_p \rho \int_{z1=-300\,m}^{z2=0\,m} T\left(z\right)dz, \tag{3}$$

where $c_p$ is the specific heat capacity of seawater (3987 $J/kg \cdot {}^\circ C$) and $\rho$ is the density of seawater (1000 kg/m$^3$). In the TOPAZ reanalysis, there are 22 vertical levels in the upper 300 m. Integration of the OHC from a depth of 300 m was chosen to ensure the incorporation of the Atlantic Water layer (AW). The intermediate depth layer, 0-300m, is characterised by the greatest AW

warming in the Arctic Ocean (Shu et al., 2022).

### 2.4 TOPAZ Reanalysis

The TOPAZ Reanalysis provides gridded data for the North Atlantic and Arctic region with a spatial resolution of 12.5 × 12.5 km (EU Copernicus Marine Service, 2024). We analysed the period 1991-2022. The dataset uses 40 vertical levels in the ocean and variables are interpolated to these depth levels. The minimum and maximum depth of the layers are 0m and 4000m

respectively. The reanalysis data is a product of the Arctic Monitoring and Forecasting Centre and contains, daily, monthly, and yearly mean fields of the following variables: temperature, salinity, sea surface height, horizontal velocity, sea ice concentration, surface heat flux and sea ice thickness. For the fields above, we use re-gridded output downloaded from Copernicus Marine Services. However, for calculating ocean heat transport as part of the ocean heat budget, we use data from the original model grid (12.5 × 12.5 km).


The dataset is based on the latest reanalysis produced by the coupled ensemble data assimilation system - TOPAZ (Xie et al. (2017) https://doi.org/10.48670/moi-00007). The TOPAZ system is based on the Hybrid Coordinate Ocean Model (HYCOM) (Bleck, 2002) coupled to an EVP sea-ice model (Drange and Simonsen, 1996). TOPAZ is forced at the ocean surface with fluxes derived from 6-hourly atmospheric fluxes from ERA5 (atmospheric reanalysis from ECMWF) (Hersbach et al., 2018).

The TOPAZ4 system uses the deterministic version of the Ensemble Kalman filter (DEnKF) (Sakov and Oke, 2008) for data assimilation. This data assimilation includes a 100-member ensemble production. Observations assimilated by TOPAZ include: SST from Operational Sea Surface Temperature and Sea Ice Analysis (OSTIA), along-track sea level anomalies from satellite altimeters, CS2SMOS ice thickness data, sea surface salinity based on the SMOS satellite (assimilated from 2013-2019), ice concentrations from OSI-SAF and in-situ temperature and salinity from hydrographic cruises and moorings collected from

main global networks (Argo, GOSUD, OceanSITES, World Ocean Database) (EU Copernicus Marine Service, 2024).

### 2.5 Observational Datasets

To assess TOPAZ's capability in representing the hydrography close to Svalbard, we compared temperature data from observations to TOPAZ output. The observational datasets used for TOPAZ evaluation are described below; satellite and mooring data were used. Mooring data was interpolated to match TOPAZ vertical levels and daily averages of the resultant time series





were obtained for plotting. For comparison with the mooring data, we chose a gridpoint from TOPAZ close to or at the location of each mooring detailed below. Temperature data from TOPAZ was compared to each mooring at the shallowest and deepest levels with sufficient valid mooring data to evaluate how TOPAZ represents the water column. The Pearson Correlation Coefficient (r) with TOPAZ data was calculated for several depths for each mooring. Where possible, at each chosen depth, we also show the correlation between monthly anomalies to ensure the correlation is not solely based on similarities in the seasonality between TOPAZ and the moorings.

### 2.5.1 NOAA Daily OISST v2.1 SST

Surface MHW events in TOPAZ were compared to NOAA Daily Optimum Interpolation Sea Surface Temperature (DOISST) data, Version 2.1 (Huang et al., 2021a; Reynolds et al., 2002) (Results, Sect. 3.2.1). DOISST provides daily SST values with a spatial grid resolution of 0.25° x 0.25°, covering September 1981 to the present. DOISST blends in-situ and bias-corrected Advanced Very High Resolution Radiometer (AVHRR) SST measurements.

### 2.5.2 Isfjorden Mouth Mooring (ISM)

Temperature measurements from an oceanographic mooring on the southern side of the Isfjorden Mouth (ISM; 78°03.660' N; 013°31.364' E, Skogseth et al. (2020), Fig. 1), were used to validate the TOPAZ model west of Svalbard. Mooring data is available from 2005-2022 and contains measurements of pressure, temperature, current velocity and salinity with a maximum depth of 240m. Data is missing for the years 2008-2010 and 2019-2020. A grid point from TOPAZ, 41.2 km offshore from the mooring, was chosen for validation with the mooring data: TP1 (78.125°N,11.75°E). TP1 is the closest point with data available at depths greater than 200m. The closest TOPAZ point to the mooring was unsuitable for comparison as it has a maximum depth of 70m.

### 2.5.3 Yermak Plateau Mooring (YPM)

An oceanographic mooring on the Yermak Plateau (YPM, Fig. 1) was used to validate the model to the north of the Svalbard West domain. The mooring was deployed at 80.118°N, 8.534°E, at 515m depth, and covers two years from August 2014 to August 2016 (Nilsen et al., 2021). The dataset contains time series of pressure, salinity, temperature and current velocity. A TOPAZ grid point at the YPM mooring location was selected for comparison.

### 2.5.4 Storfjorden Moorings, (M1,M2)

To further validate the TOPAZ reanalysis, two moorings, M1 and M2, (Vivier et al., 2019) deployed in Storfjorden were compared to the model output. The moorings were deployed as part of the STeP project (STorfjorden Polynya multidisciplinary study), a few hundred meters apart at 78°N and 20°E at a depth of 100m (Fig. 1). Data from M1 and M2 are combined to make one time series. The data used in this study covers 14 months from July 2016 to September 2017 and contains measurements





of current velocity, backscatter, salinity, temperature, and dissolved oxygen. A TOPAZ grid point at the location of the M1, M2 moorings was chosen for comparison.

### 2.5.5 Edgeøya Mooring (M4)

The M4 mooring (Kalhagen et al., 2024) was compared to TOPAZ to quantify the success of TOPAZ further east. M4 was deployed close to Edgeøya (24.407°E, 77.269°N, Fig. 1) as part of the Nansen Legacy Project. The mooring was deployed at 69m depth. The observations cover 13 months from September 2018 to November 2019. The dataset contains time series of temperature, salinity, pressure, and current velocity averaged into a common, uniform 1-hour resolution time stamp. Since M4 only provides data at the bottom, observations were compared to TOPAZ bottom temperature (60m) at the mooring location.

### 2.6 TOPAZ Evaluation

Evaluation of TOPAZ showed a strong positive correlation between daily temperatures from the ISM mooring and TOPAZ TP1 at 50m ($r = 0.8$, $p < 0.05$); Fig. A1a), and a slightly weaker correlation for monthly anomalies ($r = 0.7$, $p < 0.05$), Fig. A1b). The high correlation between TP1 and the mooring is demonstrated in Fig. A3; a moderate/strong correlation with the ISM mooring is also shown across Svalbard West. At 150m, the correlation was lower ($r = 0.6$ for daily averages and monthly anomalies, $p < 0.05$) (Fig A2a, A2b). The climatology in TOPAZ and the ISM mooring were also compared for the period 2006-2022. At 50m, there was an offset of 0.02°C between the mooring and TP1, with warmer temperatures in TOPAZ. At 150m, the offset was larger at -0.52°C with warmer temperatures shown in the mooring.

At the location of the YPM mooring, a moderate, correlation was found at 70m ($r = 0.5$, $p < 0.05$) and 500m ($r = 0.6$, $p < 0.05$) (Fig. A4). In the east, TOPAZ showed a weak, negative correlation with Storfjorden M1 and M2 moorings at 50m ($r = -0.5$, $p < 0.05$, Fig. A5) and a moderate positive correlation with the M4 mooring at a bottom depth of 60m ($r = 0.5$, $p < 0.05$) (Fig. A6). Furthermore, compared to the Storfjorden M1 and M2 moorings, TOPAZ temperatures were too low in summer/autumn. Meanwhile, compared to the M4 mooring, TOPAZ temperatures were higher than observed during summer. Additionally, at both M1-M2 and M4, during winter, TOPAZ could not resolve the cooling processes related to ice formation, and temperatures did not reach freezing as observed in the mooring data (Fig. A5, A6). Due to the short time series, monthly anomalies were not determined for these moorings. In essence, TOPAZ is shown to perform best west of Svalbard, with limitations east of Svalbard.

### 3 Results

Firstly, using the datasets detailed in the Sect. 2, this study characterises marine heatwaves (MHWs) in the Svalbard Archipelago, presenting both spatial and temporal patterns in MHWs for the period 1991-2022. Secondly, the study focuses on individual MHW events averaged over Svalbard West (Fig. 1) and examines the surface and subsurface signal of each event. Lastly, the





ocean heat content (OHC), surface heat flux (SHF) and ocean heat budget during each event are analysed to determine the
contribution of local and advective heating to the onset and maintenance of each MHW event.

### 3.1  Trends in Marine Heatwaves in the surrounding seas of Svalbard

Figure 2 depicts the mean frequency, duration, and intensity of surface MHW events for 1991-2010 and 2011-2022 around
the Svalbard Archipelago. A shift in the frequency and duration of MHW events is evident between the two periods, with a
clear increase observed in the last decade. The mean frequency and duration of MHW events for the region shown in Fig. 2,
has increased from 2 events per year and 14 days for 1991-2010 to 3 events per year and 22 days for 2011-2022. However, in
some regions, the increase in frequency is larger such as the area southwest of Svalbard (see yellow isolines in Fig. 2); here the
number of events exceeds 5 events per year. Less change has been observed in the intensity of events between the two periods;
the mean intensity during the period 2011-2022 has increased by 0.1°C compared to 1991-2010. Note that the highest MHW
intensity is located at water mass fronts, for example south-east of Svalbard at ∼74°N. High MHW intensity is also observed
on the West Spitsbergen Shelf and close to Storfjorden and the sea ice edge.



**Figure 2.** Surface mean MHW frequency (events per year), duration (days) and intensity (°C) for the period 1991-2010 and 2011-2022. The difference between the two periods is shown in the right panels. Mean TOPAZ September ice edge (sea ice concentration of 15%) for each period is indicated by the grey line. The yellow isoline represents a frequency of 5 events and a duration of 35 days. Black lines represent TOPAZ bathymetry.

To understand how MHW metrics vary on a seasonal timescale, the mean frequency, duration and intensity of surface MHW events averaged over 1991-2022 is shown for autumn (ON), winter (DJF), spring (MAM) and summer (JJAS) (Fig. 3). The frequency of surface events is shown to be highest during summer with a mean of 2 events. Meanwhile MHW frequency was



lowest in both autumn and winter with a mean of 1 event. Surface events during autumn, winter and summer are shown to be of longer duration compared to events during spring, with the longest events found in winter. The mean MHW duration for the region shown in Fig. 3 is 21 days, and 23 days in summer and autumn, and 25 days in winter, compared to 16 days in spring. MHW events with the highest intensity are found during summer - the mean intensity during summer is 0.7°C warmer compared to spring events, 0.6°C warmer compared to autumn events and 0.8°C warmer compared to winter events. In essence,

in winter and autumn there are fewer events but with longer duration and lower intensity. Meanwhile in summer there are more events but with shorter duration and higher intensity. In spring there are few events with short duration and low intensity.





**Figure 3.** Surface mean MHW duration (days) and intensity (°C) for the period 1991-2022 during Svalbard autumn (October, November) winter (December, January, February), spring (March, April, May) and summer (June, July, August, September). Mean TOPAZ sea ice edge (sea ice concentration of 15%) for each season is indicated by the grey line. The yellow isoline represents a duration of 35 days. Black lines represent TOPAZ bathymetry.





To understand in which season the changes in Fig. 2 occurred, we compared seasonal trends in MHW metrics between the periods 1991-2010 and 2011-2022. Between 1991-2010 and 2011-2022 a clear change in MHW frequency is shown in summer

with an increase of 0.2 events in 2011-2022 (Fig. A7). A change in duration is also apparent in summer with a mean increase of 8 days in 2011-2022 compared to 1991-2010 (Fig. A8). The largest change in intensity is observed in spring with a mean increase of 0.07°C in 2011-2022 (Fig. A9).

### 3.2 Characteristics of MHW events in Svalbard West

Based on Fig. 2, Svalbard West has experienced a clear increase in both the frequency and duration of MHW events and is

the region where MHW duration has increased significantly. The MHW intensity in this region is overall high compared to other regions, especially over the West Spitsbergen Shelf (Fig. 2), and is highest during summer (Fig. 3). Analysing the SST averaged over Svalbard West, we again see that the number and duration of MHWs has increased in recent decades, with a particular increase observed after 2011 (Fig. 4). Between the periods 1991-2010 and 2011-2020, for the Svalbard West region, the mean MHW frequency has increased from 2 to 3 events per year, whilst the mean MHW duration has increased from 10

days to 24 days. Little change is observed in the intensity of MHWs between the two periods with a decrease of 0.02°C in 2011-2022 compared to 1991-2010. Before 2011, MHWs are only observed in 1991, 2006, and 2007 during summer (Fig. 4). After 2011, events are shown to occur throughout the year, particularly in 2016. There is an overall increase in SST anomalies over the period of the reanalysis, and long-lasting MHW events are paired with high SST anomalies of around 2°C.



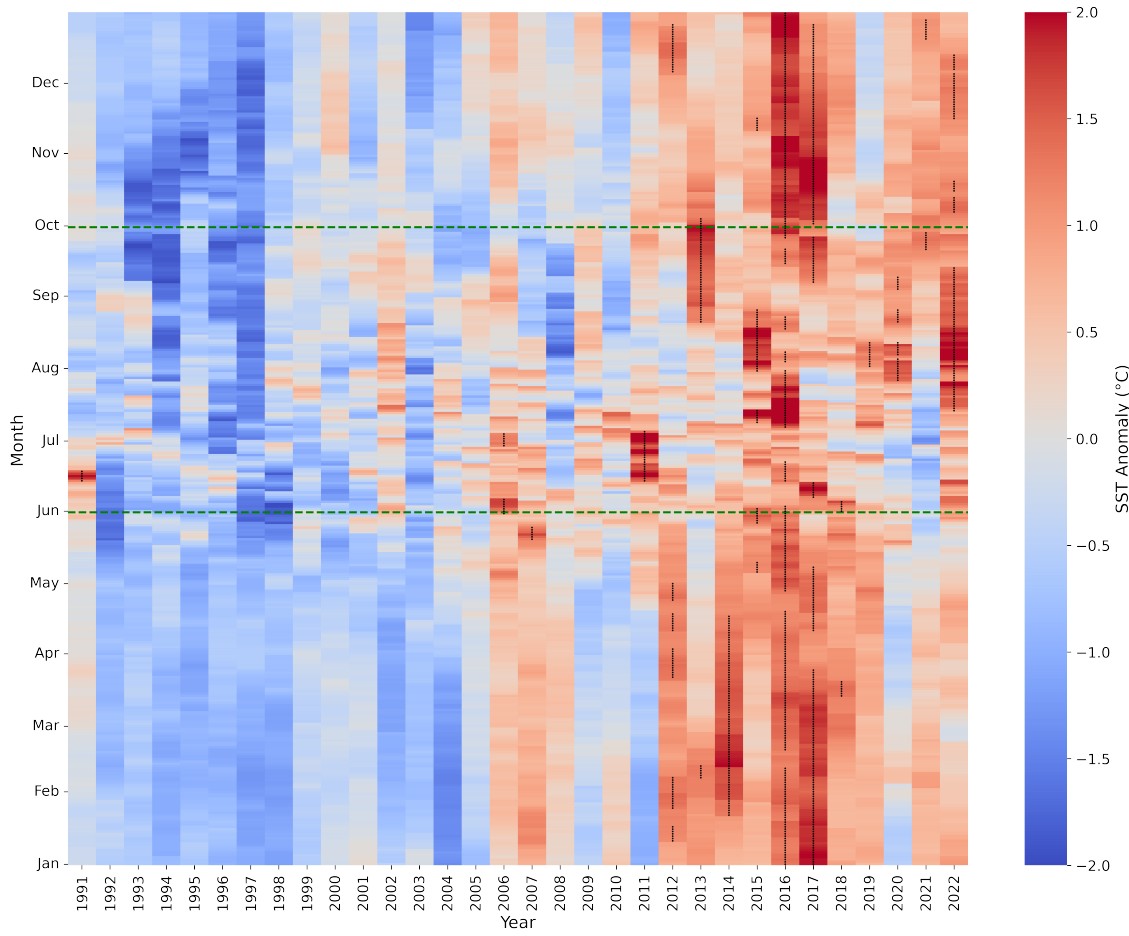

**Figure 4.** Daily average TOPAZ SST anomaly (°C) relative to the period 1991-2022 for Svalbard West (77-80°N, 5-15°E). Black dots represent when the SST exceeds the $90^{th}$ percentile, indicating the presence of a MHW. The summer period used for the focus of this study is highlighted by the green dashed line.

When the reference climatology was adjusted to the last 10 years of the reanalysis (2011–2022), the timing of most summer MHW events remained largely unchanged, whilst some (2013, 2022) were split into two shorter events, a week apart (Table A1). Furthermore, a 5–52% decrease in duration and 17–44% decrease in intensity was observed when the climatological average period was shortened. The percentage decrease in duration was larger for deep events compared to shallow events; this pattern, however, was not observed for intensity.

### 3.2.1 Horizontal Extent

Table 1 lists the summer MHW events with a minimum 10-day duration, detected using TOPAZ in Svalbard West. Each event is assigned a depth class. Four shallow ($\leq 50\,\mathrm{m}$) and four deep ($> 50\,\mathrm{m}$) MHWs have been detected. The areal coverage at the



event peak (date of max intensity) and the vertical extent for the duration of each MHW have been plotted for both shallow and deep events (Fig. 5, Fig. 6). The deep event in 2013 has the largest areal extent covering $34.7 \times 10^5 \, \text{km}^2$ at the event peak.

The shallow event in 2019 has the smallest extent covering $8.0 \times 10^5 \, \text{km}^2$ at the event peak and was largely localised to the Svalbard West region and waters southwest of Svalbard. The shallow event in 2011 was also a local event, reserved largely to the Svalbard West region, covering $8.6 \times 10^5 \, \text{km}^2$. Note that MHWs are not analysed north of the sea ice edge (sea ice concentration $\geq 15\%$).

Validation of MHW horizontal extent using DOISST satellite data revealed that the events detected using TOPAZ in Svalbard West were also present in the DOISST data (Fig. A10). Events, however, had a larger extent in the TOPAZ reanalysis (Fig. 5, 6). Compared to observations over the entire Arctic Ocean (lat $>$ 63 °N, all longitudes) TOPAZ exhibits a warm average surface bias of approximately 0.4°C during summer (Xie and Bertino, 2024). Such bias may reduce sea ice extent in the reanalysis, potentially allowing MHWs to appear in regions that should be ice-covered, likely contributing to the greater horizontal extent

observed in TOPAZ.

### 3.2.2 Vertical Extent

The maximum depth of the events in Table 1 ranged from 10-600m, with the deepest event found in 2013. Furthermore, the deep events in 2015 and 2016 have a surface event detached from a deep event (Fig. 6). In 2015, the deep event occurs towards the end of the surface event, meanwhile in 2016 the deep event persists for the entire duration of the surface event. During the

detected surface MHW events, we found temperatures in the ISM mooring to be of similar magnitude to the closest offshore TOPAZ point to the mooring (TP1) at 50m (not shown). During the detected deep MHW events, temperatures are warmer at the ISM mooring compared to TP1 at 150m (not shown); thus, TOPAZ underestimates temperatures at this depth. These findings align with the better performance of TOPAZ at shallower depths, as shown in the Methods, Sect. 2.6.





**Table 1.** Summary of summer MHW Events in Svalbard West. The Ocean Heat Content (OHC, 0-300m) is averaged over Svalbard West (SBW) for the start date of each event. Area is calculated for the event peak (date of peak intensity) within the latitude and longitude range of horizontal extent maps in Fig. 5,6.

| Event | Category | Year | Start Date | End Date | Duration | Max Intensity (°C) | Mean OHC (SBW) ($10^8$ J/m$^2$) | Area Event Peak ($10^5$ km$^2$) | Max Depth | Depth Class |
|---|---|---|---|---|---|---|---|---|---|---|
| 1 | Strong[a] | 2011 | 14-06 | 05-07 | 22 | 2.6 | 27 | 8.6 | 10 | Shallow |
| 2 | Moderate[b] | 2013* | 21-08 | 04-10 | 45 | 1.9 | 32 | 33.7 | 600 | Deep |
| 3 | Strong[b] | 2015* | 31-07 | 26-08 | 27 | 2.2 | 34 | 10.3 | 250 | Deep |
| 4 | Strong[a] | 2016* | 06-07 | 30-07 | 25 | 2.9 | 36 | 22.0 | 500 | Deep |
| 5 | Moderate[b] | 2017* | 07-09 | 26-09 | 20 | 1.8 | 45 | 18.1 | 200 | Deep |
| 6 | Moderate[b] | 2019 | 02-08 | 12-08 | 11 | 1.5 | 36 | 8.0 | 30 | Shallow |
| 7 | Moderate[b] | 2020 | 26-07 | 11-08 | 17 | 1.9 | 27 | 26.3 | 30 | Shallow |
| 8 | Strong[b] | 2022* | 14-07 | 13-09 | 62 | 2.5 | 29 | 28.1 | 50 | Shallow |

[a]Category II [b]Category I.

*Denotes years where MHWs are also detected in winter.



**Figure 5.** Horizontal (left panel) and vertical (right panel) extent of all detected **shallow** MHWS. Horizontal extent is shown for the peak date of each MHW. Vertical extent is shown for the entire MHW duration. Hatching represents where the SST or vertical temperature profile exceeds the 90[th] percentile. SST and vertical temperature anomaly (°C) are plotted in the background. TOPAZ sea ice edge (sea ice concentration of 15%) for each date is indicated by the black line in the left panel. MHWs are not detected above the sea ice edge. Green arrows (right panel) represent TOPAZ vertical levels.



**Figure 6.** Horizontal (left panel) and vertical (right panel) extent of all detected **deep** MHWS. Horizontal extent is shown for the peak date of each MHW. Vertical extent is shown for the entire MHW duration. Hatching represents where the SST or vertical temperature profile exceeds the 90$^{th}$ percentile. SST and vertical temperature anomaly (°C) are plotted in the background. TOPAZ sea ice edge (sea ice concentration of 15%) for each date is indicated by the black line in the left panel. MHWs are not detected above the sea ice edge. Green arrows (right panel) represent TOPAZ vertical levels.





## 3.3 Drivers of MHW events

To quantify whether the detected MHW events are forced at the surface by air-sea heat fluxes or forced through increased ocean heat transport (OHT), the SHF and OHT for the Svalbard West region and the OHC (0-300m) for the Svalbard Archipelago and surrounding seas (Fig. 1), have been analysed for each MHW event.

### 3.3.1 Air-Sea Interaction

The SHF summed over all grid cells in Svalbard West for the summer period of each MHW year exhibits fluctuations both above (red shading) and below (blue shading) the climatological mean (1991–2022) (Fig. 7). The climatological mean follows a clear seasonal cycle; positive SHF from June to August means heat input to the ocean from the atmosphere, whilst negative SHF from September to October means heat loss from the ocean to the atmosphere. The overall spread of SHF anomalies during the majority of the events (orange shading) remains mostly within $\pm 1$ standard deviation (grey lines). This suggests that while SHF varied during MHW events, it seldom exhibited extreme deviations from historical variability. However, in 2015, 2016, and 2020, SHF values considerably exceed the $\pm 1$ standard deviation range, particularly during the start of each MHW event. Furthermore, the long-lasting MHW in 2022 was preceded by several days where the SHF was above +1 standard deviation. Lastly, the SHF anomaly at the start of all events was positive, indicating a net heat gain in the ocean except in 2017 (event took place in late autumn and a positive SHF anomaly means less heat loss from the ocean than normal).





**Figure 7.** Total surface heat flux (W) for the summer period for each detected shallow (left column) and deep (right column) MHW event in Svalbard West. Red shading represents values above the mean (dotted line, 1991-2022), blue shading represents values below the mean. Orange shading shows the duration of each MHW in Table 1. Grey lines represent ± 1 standard deviation. The zero line (green) is shown.




### 3.3.2 Ocean Heat Budget

Prior to the individual heat budget analysis for each MHW event, we analysed the 1991-2022 annual mean time series of the ocean heat budget terms (as described in Sect. 2.2). The total cross-sectional OHT (black) oscillates between approximately 5 and 12 TW (Fig. A11). SHF (blue) remain within -6 to -12 TW, implying heat loss from the ocean surface, with a positive trend of $+0.074 \, \mathrm{TW \, yr^{-1}}$. The residual remains clustered around zero, indicating that the sum of advective transport and surface-flux terms almost account for all of the heat-budget variability. Similar analysis was then performed for each of the MHW

summer events, focusing on anomalies for each budget term. With the exception of events in 2016 and 2017 (deep events), the analysis shows that OHT anomalies exceed SHF anomalies (Fig. 8, supported by Table A2). These results indicate that OHT is the dominant driver of most events. In 2016 the event was characterised by a negative OHT anomaly, implying net heat export from the region. Lastly, OHT anomalies are considerably higher in more recent events (2019, 2020 and 2022) compared to those in 2011-2017, with anomalies clearly outside $\pm 1$ standard deviation in 2019 and 2020. SHF anomalies during the events

do not exceed $\pm$ standard deviation, consistent with results in Fig. 7.

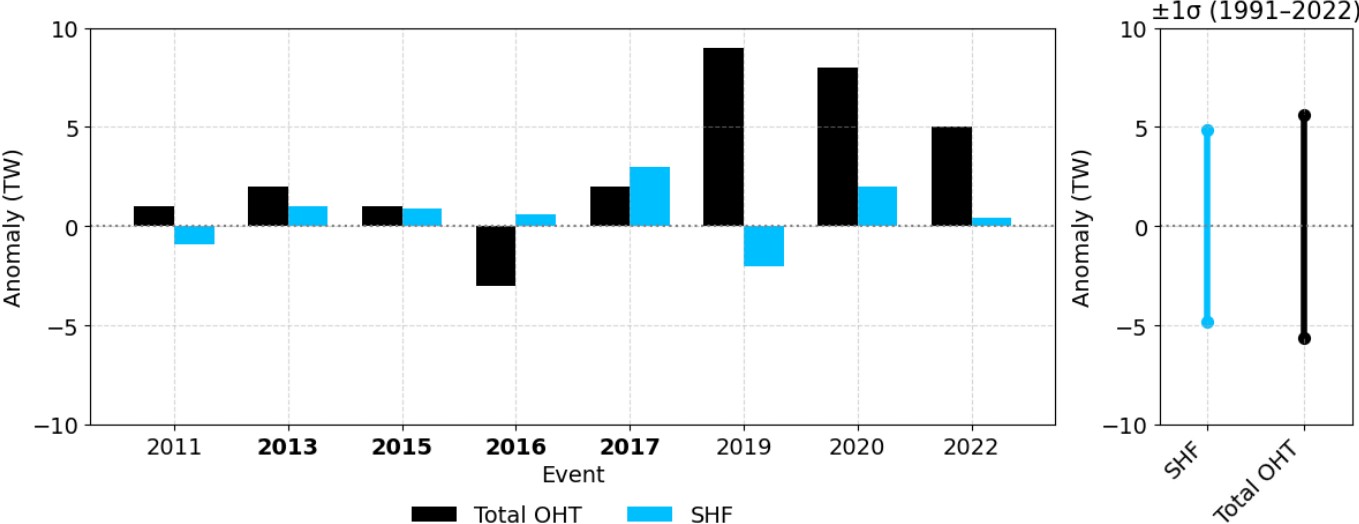

**Figure 8.** Mean anomalies (TW) of the budget terms (total OHT and SHF) for the duration of each MHW event in Svalbard West (left panel) and their standard deviation (right panel). Deep events are marked in bold. The standard deviation is calculated using summer (JJAS) daily data for 1991-2022.

Except for the deep event in 2016, all events show more heat entering through the southern boundary of Svalbard West than leaving through the northern and western boundaries – demonstrated through higher positive anomalies at the southern boundary compared to the magnitude of negative anomalies at the northern and eastern boundaries, implying an accumulation of

heat (Table A2). In 2016, the heat leaving at the northern boundary is larger than for the other events happening at a similar





time of the year. The largest heat accumulation happened in 2019 and 2020 with anomalies of 8-9 TW in the total OHT during the events. These shallow events in 2019 and 2020 have a mean positive anomaly in the western boundary implying less heat leaving at the western boundary, opposed to the other events where more heat than normal is leaving at the western boundary. Furthermore, the event in 2022 has a mean positive anomaly at the northern boundary implying less heat leaving at the northern

boundary during this event. Less heat leaving at the western/northern boundary, in addition to higher OHT across the southern boundary, could explain the high positive total OHT anomalies shown for 2019, 2020 and 2022 in Fig. 8.

Comparing deep and shallow events, deep events show small but positive SHF anomalies (0.6–3 TW) (Table A2). In contrast, shallow events exhibit both slight cooling (–0.9 to –2 TW) and warming (+0.4 to +2 TW) at the surface. Both deep and

shallow events show a significant increase in the OHT at the southern boundary (5-10 TW), where the events happening in late autumn has the largest anomalies (12 and 25 TW; the climatology is also larger).

### 3.3.3 Heat Advection

As we find an important role of OHT in driving the MHW events in Svalbard West, we further investigate the upstream OHC. Analysis of OHC anomalies (0-300m) for the start date of each event indicates that most shallow events appear to be associated

with an increase in OHC in the pathway for the Norwegian Atlantic Front Current (NwAFC, Fig. 1) (Fig. 9); the same pattern is observed in OHC anomalies integrated over 20m (not shown). Compared to the shallow events, for the deep events positive OHC anomalies were higher in Svalbard West, the Barents Sea, the Norwegian Atlantic Slope Current (NwASC, Fig. 1), and the Lofoten basin (Fig. 9).

To quantify the above difference between shallow and deep events, the OHC (0-300m) for each day in summer has been spatially averaged over regions in the NwASC pathway (Barents Sea Opening, BSO, 71–73°N, 13–22°E) and the NwAFC pathway (73–75°N, 1–10°E) (Fig. 1) during the years with prolonged summer MHW events (Fig. 10). Across both regions, OHC shows a general increase over the summer months, indicating a gradual accumulation of heat. The variability between years suggests that certain MHW years experienced significantly higher OHC than others. The bold, thicker lines (e.g., 2015

and 2016) highlight years with deep MHW events, showing relatively higher OHC compared to years with shallow events in the BSO. Meanwhile, in the NwAFC region, except for 2011, the OHC is largely higher during shallow events compared to deep events. Note, the jumps in the time series in Fig. 10 are due to the fact that data assimilation is performed weekly in TOPAZ.





**Figure 9.** Ocean Heat Content (J/m$^2$) anomalies integrated from the surface to 300m for the start dates of all detected summer MHW events. Boxes (pink - Slope Current Barents Sea Opening, BSO, 71-73°N, 13-22°E) (green - Pathway for Norwegian Atlantic Front Current, NwAFC, 73-75°N, 1-10°E) used for quantification of the impact of heat from ocean currents on each MHW are shown. Location of Svalbard West region indicated by the black box.





**Figure 10.** Summer (June–September) ocean heat content (OHC) averaged over —the Barents Sea Opening (BSO) and the pathway of the Norwegian Atlantic Front Current (NwAFC)—for years corresponding to MHW events. Dots indicate the timing of each MHW event, years with deep events are highlighted with bold, thicker lines. The locations of the regions are shown in Fig. 9.

# 4 Discussion

Marine heatwaves (MHWs) were described around Svalbard from 1991-2022 using a physical reanalysis for the North Atlantic and Arctic region – based on TOPAZ. Our analysis indicated that the duration and frequency of MHWs has increased around



Svalbard in the last decade of the reanalysis, with less change observed in the intensity of events compared to other subpolar regions (Xu et al., 2025). Focusing on summer events (June-September) in Svalbard West (77-80°N, 5-15°E) that lasted longer than 10 days, we identified the presence of four shallow ($\leq 50m$) and four deep ($> 50m$) MHWs. Through heat budget analysis, overall, we found a greater contribution of Ocean Heat Transport (OHT) than Surface Heat Flux (SHF) in driving MHW events. Our results also suggest that deep events received heat from the Norwegian Atlantic Slope Current (NwASC), whereas shallow events received heat from the Norwegian Atlantic Front Current (NwAFC).

## 4.1 MHW Definition

In this study, we applied methods from Hobday et al. (2016), whereby MHWs are detected when the daily sea surface temperature (SST) exceeds the 90[th] percentile for at least 5 consecutive days, with no more than two below-threshold days. Additionally, we used a fixed baseline of 32 years (1991-2022) to calculate the percentile. The same approach is used by Mohamed et al. (2022), studying the Barents Sea. A fixed baseline provides a stable reference for detecting long-term SST trends in MHW studies. While a fixed baseline is commonly used, some advocate for a shifting baseline to exclude the effects of climate change (Jacox, 2019). Hobday et al. (2018) and Bashiri et al. (2024), however, have advised against using a shifting baseline as updating the baseline climatology over time can change how past events are classified. Eisbrenner et al. (2024) compared a shifting baseline approach to a fixed baseline approach to study MHWs in the Barents Sea, and found that using a shifting baseline approach decreased the intensity of MHW events compared to a fixed baseline case.

Other studies on the effect of changing the baseline, for example, Lien et al. (2024), found a general trend of decreasing average intensity with decreasing length of the baseline. In this study, when the baseline was adjusted to the last 10 years of the reanalysis (2011–2022), the timing of most summer MHW events remained largely unchanged, (Table A1). However, a 5–52% decrease in duration and a 17–44% decrease in intensity was observed when the baseline was shortened. Furthermore, the decrease in duration was larger for deep events compared to shallow events. The above differences in MHW characteristics caused by changing the baseline, highlights the need, as emphasised by Amaya et al. (2023), for the definition of MHWs to be standardised.

## 4.2 Areas of high MHW activity and long-term trends

We observe regional differences in MHW activity across the Svalbard Archipelago and its surrounding seas (Fig. 2). The highest MHW intensity is located at water mass fronts. For example, south-east of Svalbard, high MHW intensity is found in the location of the Polar Front (Fig. 2, described in Mohamed et al. (2022); Barton et al. (2018)). High MHW intensity over the Polar Front could be attributed to the high variability of SST in this region (Mohamed et al., 2022). High MHW intensity, including high frequency and duration in the last decade, are also found southwest of Svalbard, along the Mohn Ridge, in the pathway of the NwAFC (Fig. 1), in the Arctic Front. The Mohn Ridge is a hotspot of sharp frontal structure and strong air-sea interaction, largely due to enhanced momentum mixing (Raj et al., 2019).



Large changes in MHW frequency and duration is found in the area close to the Mohn Ridge and also on the West Spitsbergen Shelf (Fig. 2). One reason for this could be the shoaling of Atlantic Water (AW) associated with the Atlantification seen west and north of Svalbard and in western Svalbard fjords. Atlantification refers to the increased influence of AW in the Arctic driven by recent warming of the AW inflow (Årthun et al., 2012). AW has been observed higher in the water column and along shallower isobaths in Isfjorden (Skogseth et al., 2020) and Kongsfjorden (Tverberg et al., 2019) (for location see Fig. 1). Furthermore, Strzelewicz et al. (2022) found a $8\%\,\mathrm{y}^{-1}$ increase in the volume fraction of AW on the shelf south-west of Spitsbergen. Since AW inflow along the West Spitsbergen shelf is warming (Beszczynska-Möller et al., 2012), a shoaling and increased presence of warming AW can lead to higher MHW frequency and duration.

The variability of AW temperature west of Svalbard has been shown to be associated with the strength of the Greenland Sea gyre (GSG) circulation influenced by the anomalous wind stress curl over the Nordic Seas (Chatterjee et al., 2018). A stronger GSG circulation increases the AW flow speed west of Svalbard, leading to increased oceanic heat content and higher AW temperature therein (Chatterjee et al., 2018). Thus, increased AW temperature driven by strong GSG circulation may be responsible for the increase in MHW events in Svalbard West.

## 4.3 Prolonged summer MHW events in Svalbard West

To focus on prolonged MHWs with high intensity, we examined summer (June-September) MHW events with a minimum duration of 10 days. Intense and long-duration MHWs can have a high ecosystem impact. For example, elevated ocean temperatures in summer can threaten local marine species if their thermal tolerances are exceeded (Athanase et al., 2024). When a summer MHW occurs on top of already high summer temperatures, species can be pushed towards their thermal limits, emphasising the importance of studying warm-season MHWs. Furthermore, long-term trends between the periods 1991-2010 and 2011-2022 show a clear change in MHW frequency and duration in summer (Fig. A7,A8). If these trends continue, marine ecosystems will likely face gradually increased thermal stress during the warm season. Given these intensifying patterns, understanding and predicting the impacts of future MHWs requires further particular attention to summer events.

Multiple studies use a 5-day criteria to define MHW events in the Arctic (Huang et al., 2021b; Barkhordarian et al., 2024). However, longer-duration events are shown to potentially have more significant ecological consequences. For example, a study by Dania et al. (2024) on Arctic zooplankton showed that continuous stressor exposure, to high temperatures, has a greater impact than pulse-temporal scenarios (short periods of high temperatures). In the study, continuous exposure to high temperatures for 9 days led to a more significant reduction in copepod survival compared to the pulse temporal scenario characterised by two 3-day stressor exposure phases (Dania et al., 2024). Thus, the focus should be placed on characterising and quantifying the impacts of prolonged MHW events. Such events have the potential to cause more severe and longer-lasting impacts.





## 4.4 Spatial extent of the MHW events

For each prolonged MHW event, we determined its horizontal and vertical extent. MHW events were both local (around the Svalbard Archipelago) and widespread, often extending beyond the Svalbard Archipelago (Fig. 5, 6). We also define each event based on its subsurface signal. Previous studies, for example, Zhang et al. (2023a), define the vertical structure of MHWs by
averaging temperature anomalies from in-situ profiles over the event duration. Instead of averaging, we examined the day-to-day vertical progression of each MHW by identifying the depth levels exceeding the 90th percentile on each day of the event. We found that the maximum depth of events ranged from 10-600m. Understanding the horizontal and vertical extent of individual MHW events represents a novel set of criteria that provide a valuable framework for future MHW characterisation. Understanding the vertical extent of events, in particular, can enhance our understanding of the ecosystem impacts of MHWs
by identifying which species are likely to be affected.

## 4.5 Drivers of the MHW events

### 4.5.1 Advective and Atmospheric Forcing

In previous studies, an interplay between the ocean and atmosphere has proven to be important for the onset of MHW events in the Barents Sea (Eisbrenner et al., 2024). In terms of the atmosphere, Richaud et al. (2024) found that the majority of Arctic
marine heatwaves in summer are onset by surface heat fluxes. We find positive anomalies of SHF meaning more heat input during summer, and less heat loss during autumn at the start of all events (Fig. 7). Previous studies have linked some events detected in this study to atmospheric heating. For example, Mohamed et al. (2022) suggests that warm atmospheric temperature anomalies of approximately $\geq 2\,°C$ played an important role in driving the 2016 event (Table 1) over the Barents Sea. Furthermore, Lien et al. (2024) highlights that reduced heat loss to the atmosphere during winter contributed to the onset of the
2016 event, in addition to increased inflow of AW.

Heat-budget analysis indicated that advective heating dominates in driving most of the MHWs in this study. Positive OHT anomalies exceeded SHF anomalies for all but two events (Fig. 8). Lateral advection has been highlighted as the secondary process in driving about half of the MHWs detected in the Arctic Ocean by Richaud et al. (2024). Poleward transport of heat
by boundary currents can drive subsurface heating, and through vertical mixing high SST anomalies can result in a MHW (Holbrook et al., 2019). During the majority of the detected events, the heat transport into Svalbard West across its southern boundary exceeded the combined outflow through the northern and western boundaries (Table A2). The resulting accumulation of heat can raise temperature anomalies driving MHWs.

To understand the pathways for increased OHT, we further investigated the upstream Ocean Heat Content (OHC). Our results indicated that shallow and deep MHWs were associated with the transfer of ocean heat from different sources. Our analysis revealed high OHC in the NwAFC during shallow MHWs (Fig. 10). As noted by Raj et al. (2019), the NwAFC transports warm AW poleward along the Mohn and Knipovich Ridges, feeding into the WSC and plays a key role in heat distribution





west of Svalbard, potentially driving MHW events. Meanwhile, deep events are shown to be associated with higher OHC in
the NwASC demonstrated through higher OHC in the Barents Sea Opening (Fig. 10). High heat content in the NwASC can
be responsible for the formation of deep events as the NwASC carries warm, saline AW along the continental slope at depths
between 200 and 900 m depth (strongest flow from 200-700m) (Orvik and Skagseth, 2005; Skagseth and Orvik, 2002), forming
a deep, persistent heat reservoir.

Rising temperatures in the NwASC could sustain deep MHW events by warming subsurface waters beyond the reach of surface
heating. For the event in 2016, also detected in this study, Lien et al. (2024) found the event to be preceded by increased Atlantic
Water heat transport through the Barents Sea Opening, in the pathway of the NwASC. In essence, temperatures in the NwAFC
and NwASC appear to influence shallow and deep MHW events respectively in Svalbard West, and heat transport from each
current could influence MHW frequency. Further quantification, however, of the role of each current in driving shallow and
deep MHWs in this region is required.

### 4.5.2 Sea Ice Decline

Although not addressed in this study, an additional driver of MHW events is a decline in Arctic sea ice (Huang et al., 2021b).
Previous studies have found strong feedback between sea ice cover and MHW characteristics, particularly in years with anoma-
lous sea ice cover (Mohamed et al., 2022). A decline in sea ice can increase the number of shallow MHWs, as sea ice melt
shoals the surface mixed layer, prolonging and intensifying shallow events (Richaud et al., 2024). Meanwhile, sea ice decline
could also contribute to subsurface MHW events by altering local water masses. The influence of sea ice decline on local water
masses is evident in Vivier et al. (2023). Following a winter characterised by the lowest ice coverage recorded in the Barents
Sea, observations from Storfjorden (location in Fig. 1) in July 2016 revealed that fresher surface meltwaters had been replaced
by a warmer, saltier water mass—indicating reduced vertical stratification (Vivier et al., 2023). Such displacement of fresher
cooler water masses, driven by sea ice decline, weakens stratification and promotes deeper mixing, increasing the vertical
distribution of heat, potentially driving subsurface MHWs. Reduced stratification is a key factor influencing MHW structure,
with deep events shown to occur more frequently during periods of weak stratification (Schaeffer et al., 2023). It is important
to note that Svalbard West remains largely ice-free year-round due to the influence of the WSC. However, winter ice extent in
the waters north of Svalbard has declined by approximately 10% per decade (Onarheim et al., 2014). The decline in sea ice
cover to the north may result in a decline in Arctic-origin water masses in the Svalbard West region, driving subsurface MHW
events through reduced vertical stratification.

### 4.6 Ecosystem impact of MHW events

Many Arctic species have cold and narrow thermal preferences thus appear vulnerable to MHWs (Pecuchet et al., 2025). Yet,
as emphasised in Richaud et al. (2024) and Pecuchet et al. (2025), studies quantifying the ecosystem impact of Arctic MHWs
are limited. Many studies primarily focus on the lower trophic levels. For example, Wolf et al. (2024), examined the response
of Arctic phytoplankton to heatwave conditions and found a positive productivity response. No direct studies on the impact of



MHWs on primary productivity have been completed around the Svalbard Archipelago. However, some anomalies have been reported during the MHW years we detected in this study. For example, during the timing of the event in 2011, exceptionally higher diatoms ($> 50\%$) in the phytoplankton community structure were observed both inside and outside of Kongsfjorden
(Zhang et al. (2023b), K, Fig. 1). Although not directly linked to the 2011 MHW, warmer temperatures have been shown to enhance diatom growth rates (Montagnes and Franklin, 2001). This suggests that the elevated temperatures during the 2011 MHW may have influenced phytoplankton community structure, warranting further research.

In terms of the higher trophic levels, Jordà-Molina et al. (2023) indicates the potential impact of periodic MHWs on mac-
robenthic communities in Rijpfjorden (a high-Arctic fjord in the north of Svalbard, R, Fig. 1), where a seafloor MHW in 2006 (lasting 6 days, shown in Fig. 4), resulted in a decline in species abundance and richness the following year. Lastly, our results show that west of Svalbard, MHWs can potentially reach depths of up to 600m (Fig. 6). Future studies, therefore, should additionally focus on ecosystems potentially vulnerable to subsurface heating driven by deep MHW events, such as mesopelagic and benthic organisms in the Arctic region.

**5 Conclusions**

In conclusion, an increase in marine heatwaves (MHWs) is evident in Svalbard West and around the Svalbard Archipelago in the last decade, and events are shown to be both shallow/deep and local/widespread reaching depths greater than 50m and extending from Svalbard West to the Barents Sea. Our findings indicate that compared to air-sea heat fluxes, heat advection from ocean currents plays a greater role in driving MHWs in Svalbard West. Identifying individual MHWs by their areal and
vertical extent as achieved by this study, is a useful metric that should be applied to future MHW studies to determine which ecosystems will be impacted by individual events. This study has demonstrated that ocean reanalysis, such as TOPAZ, are a useful tool for analysing the spatial extent of MHW events. As MHW research advances, greater emphasis on their ecological consequences is essential, particularly due to the fact that, driven by the climate change signal, their frequency and duration are projected to increase.

*Data availability.* All data is available for download.

**Appendix A: A1**





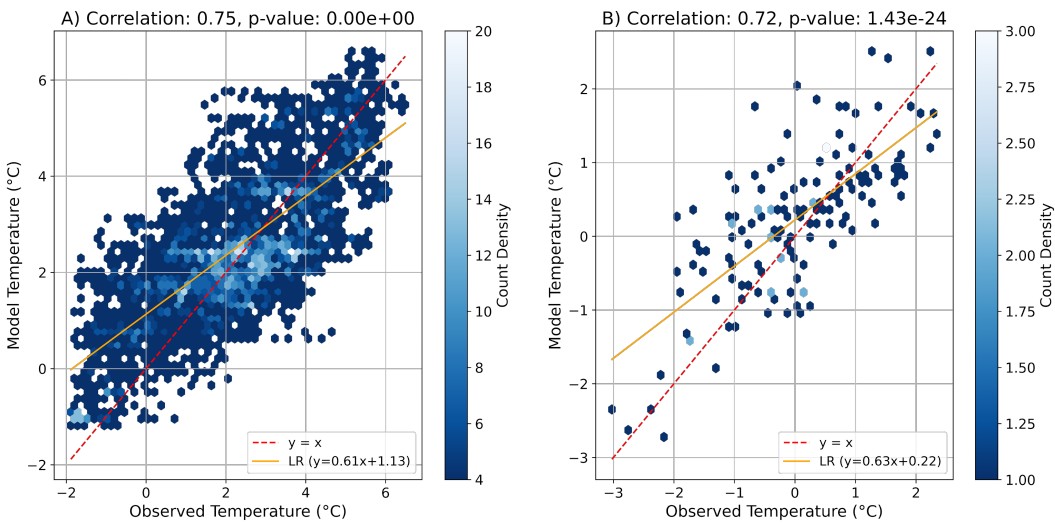

**Figure A1.** Correlation of A) daily average temperature and B) monthly temperature anomalies between TOPAZ TP1 (78.125°N,11.75°E) and the Isfjorden Mouth mooring (ISM) at **50m**. LR denotes the least squares linear regression.

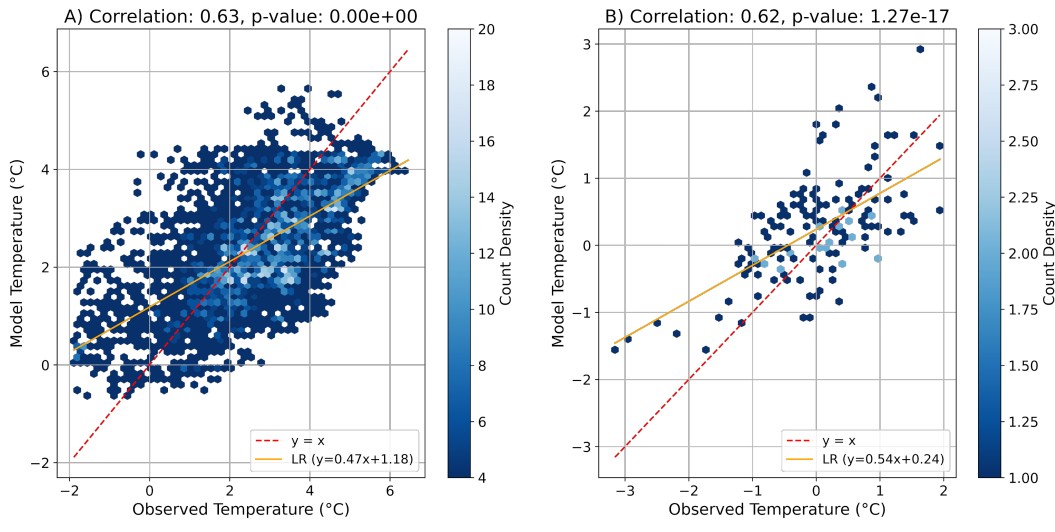

**Figure A2.** Correlation of A) daily average temperature and B) monthly temperature anomalies between TOPAZ TP1 (78.125°N,11.75°E) and the Isfjorden Mouth mooring (ISM) at **150m**. LR denotes the least squares linear regression.





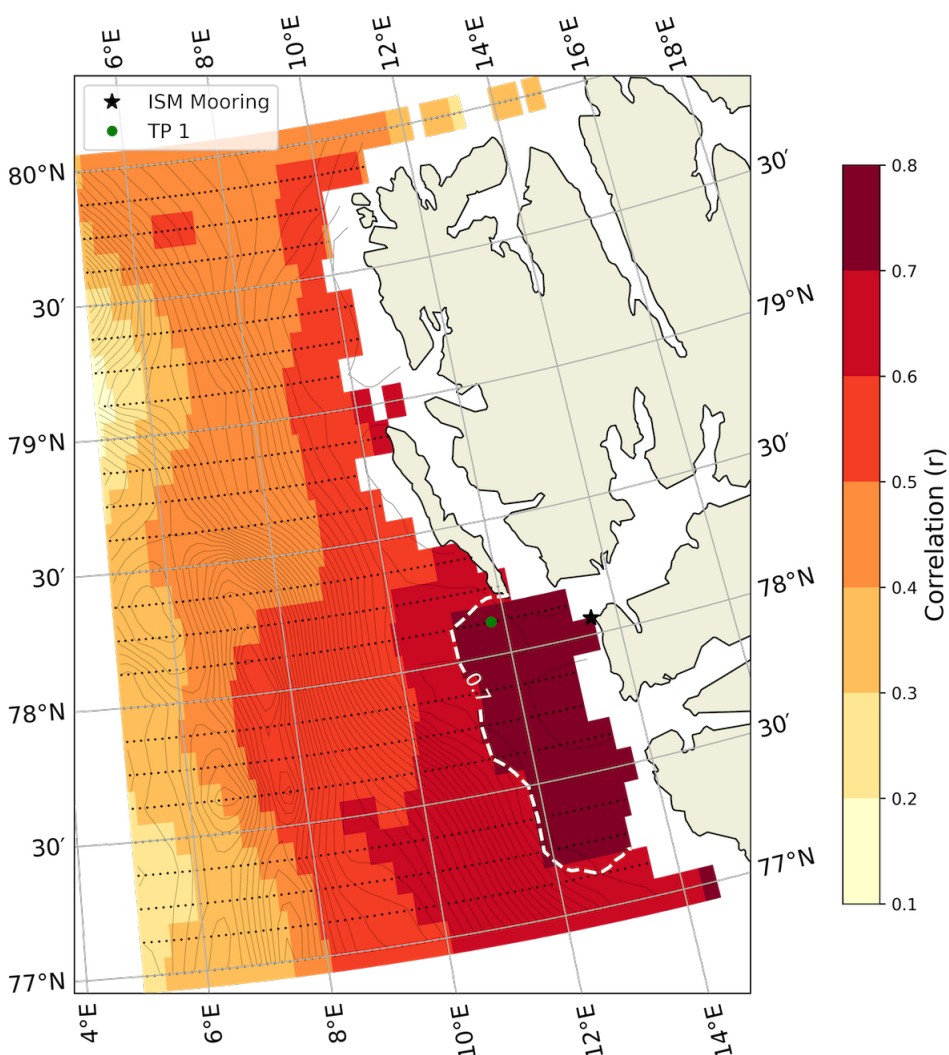

**Figure A3.** Correlation of monthly temperature anomalies between the Isfjorden Mouth mooring (ISM) and all TOPAZ grid points at **50m** in Svalbard West. Location of Isfjorden Mouth mooring and the TOPAZ comparison point (TP1) is shown. Bathymetry lines shown. Pixels with dots are statistically significant (P < 0.05). Isoline at r = 0.7 is shown.



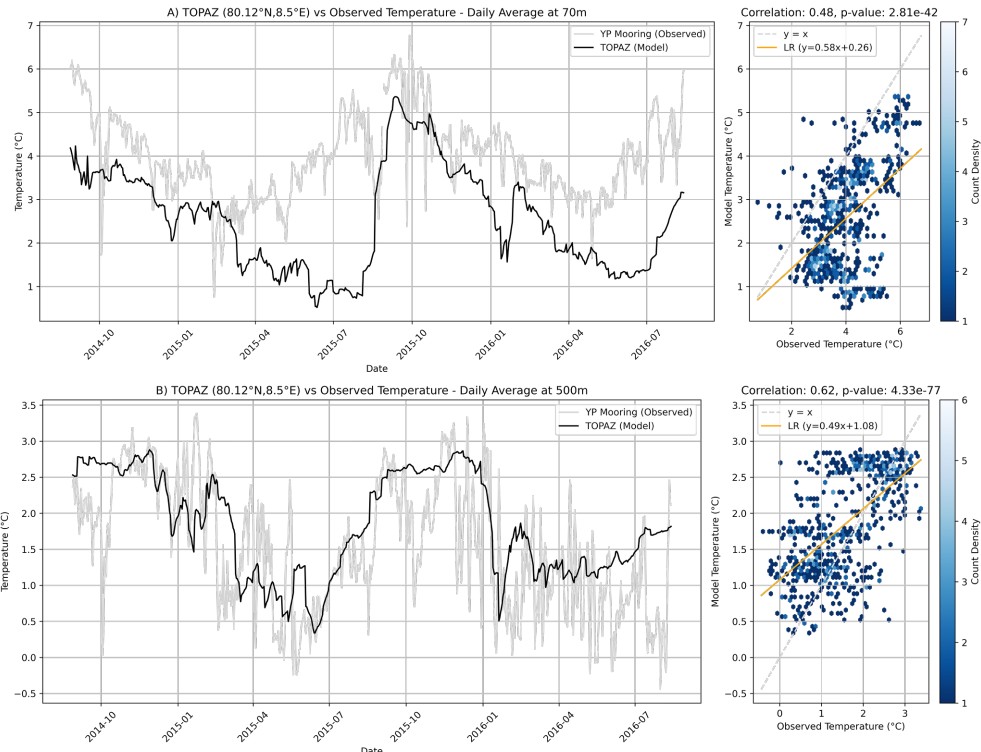

**Figure A4.** Comparison (left panel) and correlation (right panel) of daily average temperature between TOPAZ and the Yermak Plateau mooring (YPM) at (A) 70m and (B) 500m. LR denotes the least squares linear regression.

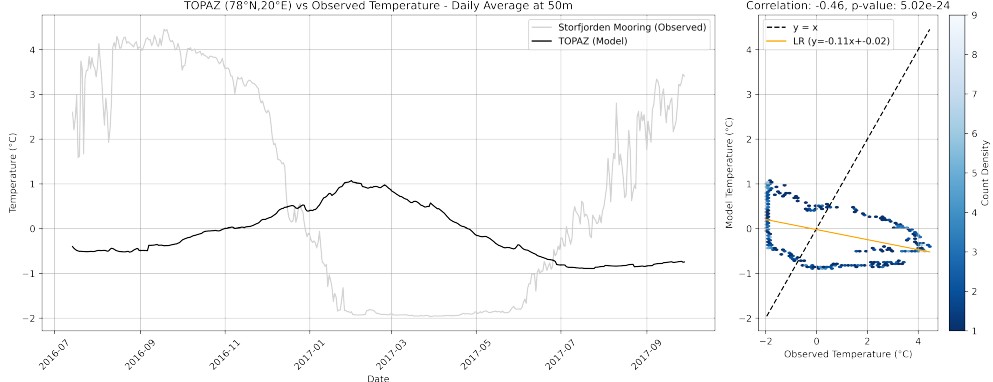

**Figure A5.** Comparison (left panel) and correlation (right panel) of daily average temperature between the Storfjorden STeP project moorings (M1, M2) and TOPAZ at 50m. LR denotes the least squares linear regression.





**Table A1.** Summary of Summer MHW Events in Svalbard West detected using a 10-year climatology (2011–2022).

| Category | Year | Start Date | End Date | Duration | Max Intensity (°C) |
|---|---|---|---|---|---|
| Strong[a] | 2011 | 14-06 | 04-07 | 21 | 2.1 |
| Moderate[b] | 2013 | 28-08 | 06-09 | 10 | 1.0 |
| Moderate[b] | 2013 | 13-09 | 01-10 | 19 | 1.0 |
| Moderate[a] | 2015 | 06-08 | 18-08 | 13 | 1.4 |
| Strong[b] | 2016 | 07-07 | 25-07 | 19 | 2.4 |
| Moderate[b] | 2017 | 17-09 | 26-09 | 10 | 1.0 |
| Moderate[b] | 2022 | 04-07 | 18-08 | 15 | 1.8 |
| Moderate[b] | 2022 | 24-08 | 06-09 | 14 | 1.0 |

[a]Category II    [b]Category I

**Table A2.** Mean heat budget terms and anomalies (shown in brackets) for the duration of each summer MHW event in Svalbard West. Surface heat flux (SHF) is summed over the area bounded by Svalbard West. Positive SHF reflects heat input to the ocean surface, negative SHF reflects heat loss. Positive ocean heat transport (OHT) indicates heat moving into the region. Negative OHT indicates heat moving out of the region. OHT values are rounded to the nearest whole number. The months for each event are listed after the year (e.g. "6/7" means the event ran from June through July).

| Event | Classification | SHF (TW) | $OHT_s$ (TW) | $OHT_n$ (TW) | $OHT_w$ (TW) | Total OHT (TW) |
|---|---|---|---|---|---|---|
| 2011-6/7 | Shallow | 3 (–0.9) | 28 (8) | –7 (–2) | –14 (–5) | 7 (1) |
| 2013-8/10 | Deep | –3 (1) | 45 (12) | –15 (–3) | –22 (–7) | 8 (2) |
| 2015-7/8 | Deep | 2 (0.9) | 35 (9) | –12 (–3) | –17 (–5) | 6 (1) |
| 2016-7 | Deep | 4 (0.6) | 29 (8) | –13 (–7) | –14 (–4) | 2 (–3) |
| 2017-9 | Deep | –3 (3) | 60 (25) | –31 (–18) | –22 (–6) | 7 (2) |
| 2019-8 | Shallow | –0.04 (–2) | 35 (10) | –13 (–5) | –7 (4) | 1 (9) |
| 2020-7/8 | Shallow | 4 (2) | 29 (5) | –9 (–1) | –7 (4) | 1 (8) |
| 2022-7/9 | Shallow | 0.6 (0.4) | 34 (8) | –8 (1) | –16 (–4) | 10 (5) |



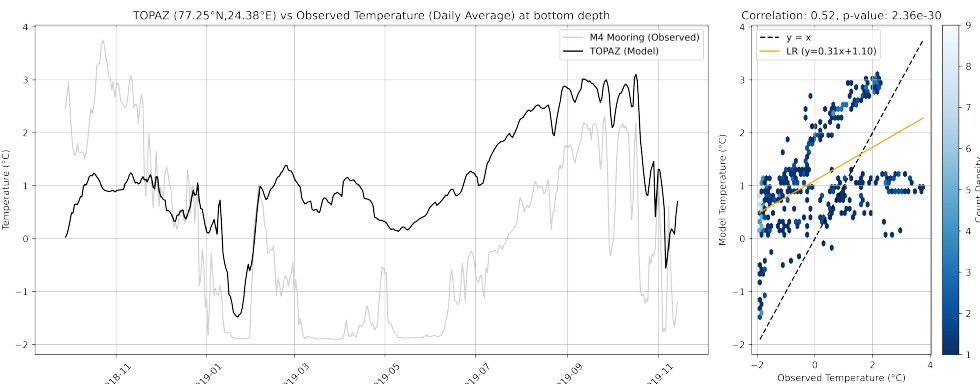

**Figure A6.** Comparison (left panel) and correlation (right panel) of daily average bottom (60m) temperature in the M4 mooring and TOPAZ at the mooring location. LR denotes the least squares linear regression.



**Figure A7.** Seasonal surface MHW, frequency (event) for the period 1991-2010 and 2011-2022 and the difference. Mean TOPAZ September ice edge (sea ice concentration of 15%) for each season of each period is indicated by the grey line. The ice edge for the difference plot represents the mean for each specific season. White space indicates that there were no MHWs events at that location.





**Figure A8.** Seasonal surface MHW, duration (days) for the period 1991-2010 and 2011-2022 and the difference. Mean TOPAZ September ice edge (sea ice concentration of 15%) for each season of each period is indicated by the grey line. The ice edge for the difference plot represents the mean for each specific season. Yellow isoline represents a duration of 35 days. White space indicates that there were no MHWs events at that location.







**Figure A9.** Seasonal surface MHW, intensity (°C) for the period 1991-2010 and 2011-2022 and the difference. Mean TOPAZ September ice edge (sea ice concentration of 15%) for each season of each period is indicated by the grey line. The ice edge for the difference plot represents the mean for each specific season. White space indicates that there were no MHWs events at that location.





**Figure A10.** Horizontal extent of all detected shallow (left panel) and deep (right panel) MHWs. Horizontal extent is shown for the peak date of each MHW using **DOISST satellite data**. Hatching represents where the SST exceeds the 90[th] percentile. DOISST SSTA (°C) is plotted in the background, and the TOPAZ sea ice edge (sea ice concentration of 15%) is indicated by the black line. MHWs are not detected above the sea ice edge.





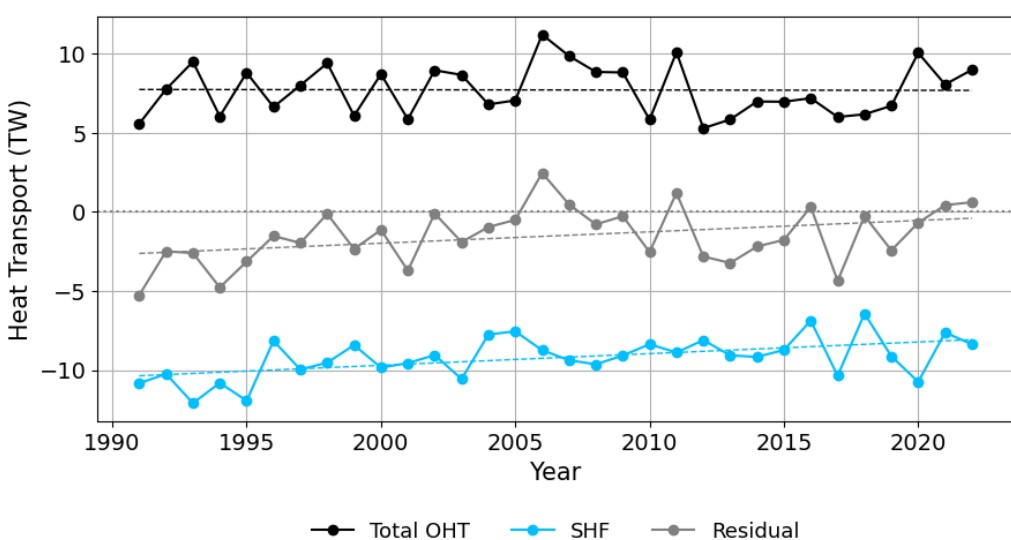

**Figure A11.** Annual time series of the Svalbard West heat budget (TW). Surface Heat Flux (SHF) is summed over the area bounded by Svalbard West.



*Author contributions.* Conceptualization: [MWK], [HRL]; Methodology: [MWK], [HRL]; Formal analysis and investigation: [MWK]; Writing - original draft preparation: [MWK]; Writing - review and editing: [HRL], [RS], [FN], [AS], [SG], [NK]; Funding acquisition: [HRL].

*Competing interests.* The authors declare that they have no conflict of interest.

*Acknowledgements.* The authors acknowledge J. Xie at NERSC for his help with the ocean heat budget analysis.

*Financial support.* MWK has an institute research fellowship (INSTSTIP) funded by the basic institutional funding through Research Council of Norway (RCN), with grant number 342603. In addition, the research leading to these results has received funding from RCN through Climate Futures (grant 309562), MAPARC (grant 328943), and from the Nansen Center institutional basic funding (RCN grant 342624).



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
