# Peer review of "Characterising Marine Heatwaves in the Svalbard Archipelago and Surrounding Seas"

_EGUsphere, 2025_

## Author Comment (AC1)

**Reviewer 1**

Review comments for "Characterising Marine Heatwaves in the Svalbard Archipelago and Surrounding Seas" by Williams-Kerslake et al.

The authors presented a comprehensive MHW study in the Svalbard Archipelago region using TOPAZ analysis, which is validated by various moorings and OISST analysis. They presented MHWs changes in timescales of decade and season, vertical and horizontal extent, provided heat budget analysis for each MHW events, and concluded that the most MHWs are driven by the ocean heat transport. The manuscript is well written and can be published in EGUsphere after revision. My major concern is what drive the deep MHW, if the surface heat flux, how?

We thank the reviewer for their positive comments and constructive feedback. We provide detailed responses to each comment below and will revise the manuscript accordingly. With regards to the reviewer's major concern, we find that both surface heat flux and ocean heat transport contribute to the development of deep events. We will review the text to make this finding clearer.

1. L54: Use consistent time unit in L54 °C year-1, L57 %y-1, L62 °C per decade

We will change the units to be consistent.

2. L110, 90th percentile. I am not sure whether the region is ice free during the summer from 1991-2022. If not, how the MHWs are defined in the ice-covered region, since water temperature changes a lot when ice is melted. E.g. the threshold, which is calculated using the temperature with ice in the early period, may be difficult to applied to the time when ice in melted in the later period. Can you test how much MHW features are changed if the threshold is set to 95th percentile?

Svalbard West has been ice-free during the summer from 1991 to 2022, so testing with different thresholds is not relevant in this context.

3. L128, equation (2), why is Tref is needed?

Tref is 0°C, therefore this could be removed from the equation.

4. Figures A1B, A2B and other figures with p-value: 1.43e-24, 1.27e-17, check and revise.

We believe the low p-values are due to a high number of observations. We will check each p-value.

5. L211-219, Figs. A5 (low correlation) and A6 (bifurcated correlation), these figures may indicate the biases of TOPAZ model in the coastal regions.

Yes, we agree. TOPAZ does not perform well on the coast at the location of Mooring M1-M2 and M4, as it is unable to resolve the cooling processes related to ice formation (L216-217). Storfjorden (M1-M2) and M4 experience intense water mass transformation due to sea ice

freezing, and the Storfjorden mooring is situated in a productive polynya. TOPAZ does, however, perform well on the west coast of Svalbard, as shown by the comparison with the Isfjorden Mouth Mooring (L203-209). TOPAZ effectively resolves the bathymetry of the Isfjorden trough, which is why we see a high correlation with the Isfjorden Mouth Mooring. We will add some clarification to the text to explain this difference.

6. L234, note that the intensity decreases in many regions although frequency and duration increase in most of the regions. Therefore, it might be helpful to use the cumulative intensity by integrating SSTA and time in units of degree-day (e.g. Huang et al. 2025, DOI 10.1175/BAMS-D-24-0337.1), which will enable us to see how MHWs intensify with time.

Thank you for the suggestion. We will add cumulative intensity to Figure R1 (below) to show how MHWs intensify with time.

7. L240-246, it might be helpful to add implications or causes of those features, e.g. warming is strong in winter than summer etc.

We are unsure what is meant here. We find that the intensity is higher in summer than in winter, but the duration is shorter, so the accumulated intensity may be larger in winter than in summer. This point is interesting; we have, however, not investigated this. Our aim was not to analyse seasonal variations in sea surface temperature (SST), but rather to compare summer MHW metrics with those of other seasons, given our focus on summer MHWs. Therefore, we believe it is best to not include this in the text, and this could be investigated in a later study.

8. L248-252, The definition of MHW differences is not straightforward: there are many regions without MHWs in left panels marked as "missing", which results the difference in right panel are marked as "missing" or blank. Can the "missing" in the left panels be marked as "zero"? This should make the difference more reasonable. One alternative way is to assess the differences is to integrate MHWs in space and then compare their time evolution.

Thank you, we agree that the definition of MHW differences is not straightforward. Instead, as suggested, we will integrate MHWs in space for each season and then compare their time evolution. We will also add cumulative intensity to this plot. Please find the plot below:

Figure R1: Spatially averaged MHW frequency (events), duration (days), intensity (°C) and cumulative intensity (°C days) for the Svalbard Archipelago and surrounding seas (69-82°N, -10-35°E) for Autumn (ON), Winter (DJF), Spring (MAM) and Summer (JJAS). Data is smoothed using a 5-year running mean. MHWs are not analysed north of the sea ice edge (sea ice concentration ≥ 15%). Dashed lines represent the linear fittings.

9. L279, "Note that MHWs are not analysed north of the sea ice edge (sea ice concentration  $\geq$  15%)." This might be noted much earlier in definition in section 2.1.

Thank you for your comment. We will add "To generate maps of MHW metrics in Svalbard West and surrounding seas, MHWs were also detected individually for each grid point in the seas surrounding Svalbard. For this analysis, MHWs were not analysed north of the sea ice edge (sea ice concentration  $\geq 15\%$ )" in Section 2.1.

10. Figures 5, 6, "peak date of each MHW". How is this defined? MHW evolution may not be synchronized in different regions, and therefore it is not straightforward to define "one" MHW within a large region (more than one grid point). What the black dots represent?

The peak date of each MHW is the date of peak intensity (maximum SSTA) for the MHW events detected using SST averaged over Svalbard West. For the peak date for each Svalbard West event, we then see in the larger mapped area (Figures 5, 6) if the grid cells exceed the 90th percentile. We understand the limitations of basing the peak date for the larger area on the peak date taken from the spatial average of Svalbard West, as the peak date is likely to differ for each grid cell. However, we chose this method since the MHW events we identify are based on Svalbard West. We will clarify this in the text.

11. L315, 332, "With the exception of events in 2016 and 2017 (deep events)," Does this imply that the deep MHWs are driven by the surface heat flux, which is hard to imagine. If not, what drive the deep MHWs? "With the exception of events in 2016 and 2017 (deep events)," why?

In 2017, the surface heat flux (SHF) anomaly is only slightly larger than the ocean heat transport (OHT) anomaly, suggesting that both SHF and OHT contributed to the event, with SHF playing the stronger role. In 2016, despite a net negative OHT anomaly driven by anomalous heat export at the northern boundary, an 8-TW positive anomaly still entered the region through the southern boundary. Thus, even though substantial heat was lost overall, the anomalous heat import—together with SHF—could have contributed to the development of the MHW event. Thus, the deep events are not solely driven by the surface heat flux. We will make this finding clearer in the text.

**12. Figure 10, suggest exchange the dotted with solid lines, which will highlight the MHWs.**

As mentioned by Reviewer 2, this section on the impact of heat advection on MHW events requires further analysis for us to be able to make concrete conclusions. As a result, we have decided to remove this section.

**13. Section 4, Discussion, the discussion is lengthy and should be shortened.**

Thank you. We agree that the discussion needs to be shortened. The section on heat advection will be removed since this will no longer appear in the results. We will also remove the section on sea ice decline since our results do not focus on sea ice. Additionally, we will shorten the section on the ecosystem impact of MHW events. We have also noted that there is a lot of repetition of the results and methods in the discussion. We will move any methodological justifications to the methods section. We will also remove any repetition of how analyses were performed, specific numerical values, figure references that were already presented in the results and detailed descriptions of individual events, unless they are directly needed for interpretation. This will ensure that the discussion only focuses on the main findings and how they fit into the broader context.

---

## Author Comment (AC2)

**Reviewer 2**

**Review comments for "Characterising Marine Heatwaves in the Svalbard Archipelago and Surrounding Seas" by Williams-Kerslake et al.**

This study provides an overview of marine heatwave (MHW) characteristics around the Svalbard Archipelago in the Eurasian Arctic Ocean, using the regional ocean reanalysis TOPAZ over 1991-2022. This work covers MHW seasonality, long-term trends, and horizontal and vertical extent, and investigates the relative roles of surface heat fluxes and ocean heat transport in driving the MHW events. The article presents two very valuable aspects. First, MHW studies in the Arctic region still remain limited, and the present work constitutes a nice addition. Second, the authors characterise the vertical structure of MHWs, which is still rarely done—and should be more often, in my view. Therefore, I find the focus of this paper important and timely. The approach appears generally robust.

However, the manuscript currently presents some shortcomings, mainly a lack of detail in the methodology employed and insufficiently supported interpretations. Still, I believe these points can be addressed and that after revisions, this study will constitute a very valuable contribution to the journal and the MHW research field. My suggestions are detailed below, listing first my major/general comments, and more minor typesetting points thereafter.

We thank the reviewer for their positive comments and constructive feedback. We provide detailed responses to each comment below and will revise the manuscript accordingly.

**Main comments:**

- 1. The introduction is lacking several key MHW references, both from literature covering global MHW properties and trends, and from the limited literature on Arctic MHWs. This particularly applies to paragraphs lines 14-21 and 33-41. Below are examples of very relevant papers for the authors to include (either all or a selection of them):
  - Capotondi et al. (2024), A global overview of marine heatwaves in a changing climate. 10.1038/s43247-024-01806-9
  - Malan et al. (2025), Lifting the lid on marine heatwaves. 10.1016/j.pocean.2025.103539
  - Hu et al. (2020), Marine heatwaves in the Arctic region: Variation in different ice covers. 1029/2020GL089329
  - Golubeva, et al. (2021), Marine heatwaves in Siberian Arctic seas and adjacent region. 10.3390/rs13214436
  - Mohamed et al. (2022), Marine Heatwaves Characteristics in the Barents Sea Based on High Resolution Satellite Data (1982–2020). 10.3389/fmars.2022.821646
  - He et al. (2024), Arctic Amplification of marine heatwaves under global warming. 10.1038/s41467-024-52760-1
  - Léon-FonFay et al. (2024), Sensitivity of Arctic marine heatwaves to half-a-degree increase in global warming: 10-fold frequency increase and 15-fold extreme intensity likelihood. 10.1088/1748-9326/ada029

• Gou et al. (2025), The changing nature of future Arctic marine heatwaves and its potential impacts on the eco-system.10.1038/s41558-024-02224-7

We thank the reviewer for noting these important studies. We will add these studies into the introduction.

- 2. Some more details on the methodology could be needed. For example:
- A) Could you provide the reason for choosing the period "cut-off" in 2010-2011 as soon as it is introduced? Otherwise, this leaves the readers to wonder until rather late in the manuscript.

At first, we compared maps of MHW characteristics every 10 years and found that the interdecadal variations in MHW characteristics between 1991-2001 and 2001-2010 were negligible, so the first two decades were merged into a single period. After 2010, a sudden increase in MHW frequency and duration occurs, which is why we focus on this period. We will add clarification to the results text.

B) What is the reasoning to pick a different split for the seasons than meteorological? I would find it OK, but it is good to provide a rational to readers and/or to explain how routine this is in the field.

The season split we chose represents the warmest (summer) and coldest (winter) months in the ocean around Svalbard and then shoulder seasons (fall and spring). A similar seasonal partitioning was used by Schlichtholz et al., (2021) in the Barents Sea. Summer in Svalbard's Isfjorden is also defined as JJAS by Skogseth et al., (2020). We will add this rationale to the text.

C) How are horizontal bounds defined for each event? I.e. how do you detect coherent events/patterns of MHW?

Each event (in Table 1) is detected using sea surface temperature (SST) averaged over Svalbard West (77-80°N, 5-15°E). For the maps of MHW characteristics, the MHWs are detected for each grid cell in the latitude-longitude range 69-82°N, -10-35°E. We also determine the horizontal extent of each MHW on the peak date of the event. The peak date of each MHW is the date of peak intensity (maximum SSTA) for the MHW events detected using SST averaged over Svalbard West. For the peak date for each Svalbard West event, we then see in the larger mapped area (Figures 5, 6, 69-82°N, -10-35°E) if the grid cells exceed the 90th percentile, excluding data above the sea ice edge (sea ice concentration ≥ 15%). The area (km²) of each event in Table 1 is thus based on the region 69-82°N, -10-35°E. We will re-order the results text to make it clearer whether the metrics we describe are based on the Svalbard West average or per grid cell for a larger area.

3. Part of the interpretations are not sufficiently or not clearly enough supported by analyses (or past literature, if relevant). It would be good to support interpretations of causal

relationships with quantitative analyses & to directly illustrate them with corresponding figures. Importantly, the authors conclude that shallow and deep MHWs were associated with the transfer of ocean heat from different sources, with more heat from the NwAFC during shallow MHWs, and more from the NwASC for deep MHWs (l. 345-353 and l. 450-459). However, I don't see a particularly clear separation between shallow and deep on Fig. 10. Given your small event sample size, is this separation robust? Could you provide more quantitative estimates for this?

We found the results for this part interesting; however, we agree that a larger sample size and more analysis is needed to conclude on the role that the heat transport of these two currents play in deep and shallow MHWs. We will therefore remove this section. This can be followed up in a later study.

4. How do the present findings of a dominant role of OHT around Svalbard fit with previous studies finding that surface MHWs in the Arctic are primarily driven by SHF? These are not necessarily mutually exclusive, but could you include some discussion on this point?

Thank you, a very interesting point. We will add a paragraph in the discussion on this. One study that we will use to describe this difference is Richaud et al. (2024), which describes that ocean heat transport can become the main contributor to MHWs at the main Arctic gateways. We will also refer to Lien et al. (2024), which mentions that Atlantic Water transport into the Barents Sea through the Barents Sea Opening was high before the 2016 MHW.

5. While the manuscript is generally clear enough to convey its message, I would still suggest added attention to improve the writing flow, and double check grammar if possible. There are some redundancies, occurrences of colloquial phrasing, and typesetting issues (unnecessary spaces, hyphens, or commas, and occasionally text/information that has been inserted twice in a row). I also know that paraphrasing findings or interpretations from previous studies can sometimes feel a little artificial. However, I noted a few cases of very direct usage of the complete phrasing appearing in past studies, and that, not just in the introduction but also when discussing results. Examples can be found e.g., 1. 404-409 and 1. 415-419 for the respective cited papers. I would encourage the authors to carefully review their phrasing, to avoid such re-uses and rather foster their own original formulations, wherever possible.

Thank you, we will reduce redundancies in the text, check grammar and remove typesetting issues. We will also carefully review our phrasing of results from other studies.

**Minor comments & edits:**

6. General: There are quite a lot of appendix figures. Are all panels absolutely essential to supporting the analyses? Would it be possible to reduce that appendix figure number by merging some together and/or by including key information in the main figures? This may lighten the overall manuscript a little, but this is not absolutely essential.

Thank you, we agree that there are a lot of appendix figures. To solve this, we will remove Figure A11 and instead quote the values from this figure in the text. We will also merge Figures A7, A8, and A9. As suggested by Reviewer 1, these figures will now be one 2 x 2 figure showing the mean time series for frequency, duration, intensity, and cumulative intensity averaged over the whole map region shown in Figures A7, A8, and A9 for each season.

7. L. 52-54: Can you include a citation to support this sentence? 'Warming AW and a rise in regional sea temperatures, intensified by the loss of sea ice, are potential reasons for the high frequency of MHWs around Svalbard.'

This sentence was our own hypothesis, so we will instead change this paragraph to a general description of how the climate has changed in the Svalbard Archipelago with mention to references.

8. Fig. 1: Could we have some indication of the depth(s) of these currents? Rather surface / subsurface? I know the AW is in some places at the surface, in some subsurface. It could be good to indicate what is where somehow (e.g. dashed or thicker lines for subsurface). Also, please capitalise "mooring" consistently here.

We will include the depth range of AW flowing past Svalbard in the introduction. We will also ensure mooring is capitalised consistently. In Figure 1, we will indicate with a darkening of the red arrows where AW becomes gradually more subsurface (north of Kongsfjorden latitude).

9. L. 86: "no more than two days below the threshold" could you justify why adding this somehow more relaxed condition?

This condition is part of the definition by Hobday et al. (2016), which states that "gaps between events of two days or less with subsequent five-day or more events will be considered as a continuous event". The same method is applied in Mohamed et al. (2022), which states that "two consecutive MHW events with a gap of 2 days or less are considered a single event".

10. L. 99: This is interesting. So, this may oversee events that do not have a surface expression, right? That is OK, just needs to be stated for clarity (and discussed in the discussion) –

Yes, this is correct. Our study is overlooking events that do not have a surface expression. We will state this in the methods section for clarity. We will also discuss this in the discussion with reference to Malan et al. (2025), which classifies subsurface MHW events, and Lien et al. (2024), which identifies bottom MHWs in the Barents Sea.

11. L. 106: How are anomalies computed (for each data product)? Against which reference period?

Anomalies are calculated relative to the reference period 1991-2022. We will add this clarification to the methods section.

12. L. 115. What do you mean by "unique identified"?

By 'unique identifier' we mean category. We will replace this phrase with the word 'category' for clarity.

13. L. 123. Rho should be rho\_0?

Thank you, yes, we will change this.

14. L. 129: " $z(\lambda)$  is the depth at each section", meaning, down to the ocean floor?

Thank you, yes, we will clarify this in the text.

15. L. 204: Here and throughout the manuscript, please remove parentheses after figure number/subpanels, when quoting in text.

We will review the text and remove this typesetting error.

16. L. 204: "...r = 0.8, p <0.05); Fig. A1a), and a slightly weaker correlation for monthly anomalies (r = 0.7, p <0.05)," Please round values consistently between text & figure (here the figures say 0.75 and 0.72)

We will round values consistently between the text and figures.

17. L. 234: "...the highest MHW intensity is located at water mass fronts, for example southeast of Svalbard at~74°N" Could you tell readers to which water mass front this corresponds? Same for subsequent examples.

L234 corresponds to the Polar Front. We will add this into the text.

18. L. 236-239: This is interesting. However, this should not be in a section titled "trends". Please restructure or retitle accordingly. e.g., Seasonal variations and trends (with, in order, seasonal variations, annual trends, seasonal trends).

Thank you, we will rename this section to 'Seasonal variations and trends' and follow your suggested order.

19. L. 239: To match the colour scale on the figure, might be best to say 1-to-3 events in summer, vs. 1-to-2 events in other seasons.

Since all seasons show 1-2 events, we find it best to stick with the mean frequency value for each season averaged over the map region in Figure 2 (69-82°N, -10-35°E, excluding data above the sea ice edge).

20. L. 242: "the region shown in Fig. 3" So, the whole map shown? or the black box?

The whole map shown. We will clarify this in the text.

21. L. 243-244: This is a strange formulation. Why not give the intensity amplitude directly for all, as shown in the figure?

We agree. We will change the text to instead give the intensity amplitude for all seasons.

22. L. 251: "The largest change in intensity is observed in spring" Interestingly, I see rather that large parts of the WSC experience a decrease. Intensity changes in summer and spring vary also quite a bit spatially, so it would be good to provide a bit more understanding of why or at least propose reasons. And is this averaged on the box? or on the whole figure? This would need to be a bit more precisely described here.

The decrease in the WSC is very small and we are unsure why this occurs. Values are averages of the whole figure. We will replace this figure with a time series of mean MHW metrics for the mapped area for each season as suggested by Reviewer 1.

23. Figure 2: this is a nice visual, but it is hard to see whether there is indeed an abrupt change in absolute intensity or whether this is a visual artifact from the colourbar and/or the fact that we are looking at anomalies here (so, bound to switch from blue negatives to red positives once the transient warming exceeds the 30-year climatological reference period).

Thank you, however, we are a bit unsure what is meant by this comment. We compare the 2011-2022 average with the 1991-2010 average, not the 30-year climatology. The first two decades showed a very similar pattern, hence we averaged over these two decades and compared the results to the last decade. Therefore, the difference is the change of the last decade relative to the first two decades.

24. L. 254-255: the same information appears twice in this sentence.

Thank you, we will remove this redundancy.

25. L. 256-257: Again, this is redundant with the first sentence, and the third time this information appears in this paragraph.

L256-257 refers to SST averaged over the Svalbard West region, whereas before (L254-256), the text refers to MHW metrics per grid cell in the Svalbard West region. We will clarify this difference in the text.

26. L. 265-269: "a 5-52% decrease in duration and 17-44% decrease in intensity" Is this a trend, a reduction of the trend, or a reduction in mean value? Where are the ranges coming from? What is the reason behind testing a shorter reference period for anomaly computation? A lot more information is needed here to understand how these results are obtained and their physical meaning.

Thank you, these results will be better explained in the revised manuscript, as suggested below:

The sensitivity of MHW detection to the choice of reference period is well documented in literature (e.g., Smith et al., 2025; Lien et al., 2024). To assess this sensitivity, we examined how the MHW metrics for each event in Table 1 changed when using a shorter climatology. When the reference period was restricted to the final 10 years of the reanalysis (2011–2022), the timing of most summer MHWs remained similar. However, a few events (2013, 2022) were divided into two shorter events occurring about a week apart (Table A1).

For events whose timing was largely unaffected by the change in climatology (i.e., those with matching dates under both the 1991–2022 and 2011–2022 baselines), we compared their duration and intensity. Under the shorter climatology, event duration decreased by 5–52 % and intensity decreased by 17–44 %. The percentage range shows the smallest to largest decrease in mean duration and intensity under the 2011–2022 baseline relative to the 1991–2022 climatology.

In summary, shortening the climatology does not substantially alter the timing of most MHWs, but it does reduce their intensity and duration.

27. L. 270: This section should probably be merged into one subsection "horizontal and vertical extents"

Thank you for your comment. However, since the vertical extent is based on the mean temperature profile for Svalbard West during the entire duration of each event, whereas the horizontal extent is determined for each grid cell in a larger area at the peak date of each event, we find it best to keep these sections separate. We will clarify this in the manuscript.

28. L. 299-309: In this section, could you provide more quantitative information? E.g. line 305, "SHF values considerably exceed the  $\pm 1$  standard deviation range": How many std variation? for how long/how frequent? Otherwise, I would suggest shortening this part.

Thank you, we will add in some more quantitative information including by how many standard deviation values exceeded the  $\pm 1$  standard deviation range and for how long/frequent. We will also shorten this part.

29. Figure 8: which kind of STD is this? (spatial, across events...)

The standard deviation was calculated from Svalbard West mean daily sea surface temperature for all summers (JJAS) from 1991-2022. We will clarify this in the text.

30. Figure A2: Define your OHT "s, n, w" explicitly (I assumed it is through each bound of the box)

Thank you. It is through each bound of the Svalbard West box. We will clarify this in the table caption.

31. L. 313-315 and Figure A11: There is no residual term in your heat content tendency equation, so can you please explain how this quantity was calculated in your figure A11?

Figure A11 has now been removed from the manuscript to reduce the number of appendix figures. The residual is the difference between ocean heat transport (OHT) and surface heat flux (SHF) terms.

32. L. 325-331: which figure(s) supports this analysis here? Is it still based on Table A2? If yes, I do not see how the 2016 event has a larger northward transport compared to 2015, 2017 and 2019. The proposed causal relationships could be supported more directly with quantitative analyses & directly illustrated by figures.

Table A2 supports this analysis here. 2016 (occurs in July) has a larger northward transport anomaly (OHTn, Table A2) compared to events happening at a similar time of year; for example, when compared to 2015 (occurs July-August) and 2020 (occurs July-August). We make this inference due to the impact of season on ocean heat transport (i.e., more heat transport at the end of summer). To enable the reader to compare the OHT values by season, we will sort the event column in Table A2 based on season.

33. L. 325: "implying an accumulation of heat", could you elaborate on what is meant by this? By which processes would heat accumulate there? Could it also imply a possible enhanced heat loss from the ocean to the atmosphere, or a larger heat flux to the Svalbard shelf/coast?

By heat accumulation we mean the imbalance between heat entering and leaving the Svalbard West region through the oceanic boundaries (OHTs, OHTn, OHTw). For this result we will make sure to refer to the mean net OHT at each boundary. Through analysis of mean net OHT values at each oceanic boundary, we find less heat leaving the Svalbard West region than entering it, resulting in a net retention of heat within the region. We will then look at anomalies of OHT at each oceanic boundary to see how they compare to anomalies in SHF. The text will be re-written to clarify our results. Furthermore, to more clearly illustrate how the OHT varies at each oceanic boundary of Svalbard West during each MHW we will include a new panel in Figure 8 that presents the mean OHT anomaly at each boundary (Figure R2a). In terms of whether these results could imply a larger heat flux to the Svalbard shelf/coast, the boundary for Svalbard West for our OHT calculations is at the coast therefore, any heat advected toward the coast remains within the defined Svalbard West analysis region and does not leave this box.

Figure R2: A) Mean OHT anomalies at the southern (OHTs), northern (OHTn), and western boundary (OHTw), of Svalbard West during each event. Deep events are marked in bold. B) Mean anomalies (TW) of the budget terms (total OHT and SHF) for the duration of each MHW event in Svalbard West (left panel) and their standard deviation (right panel). The standard deviation was calculated from Svalbard West mean daily sea surface temperature for all summers (JJAS) from 1991-2022.

34. L. 339: is a single-day snapshot really representative here? Would it be possible to average over a coherent time-period of days to weeks to give more robustness to your results?

As discussed in our response to comment 3 (see above), this section will be removed.

35. L. 339-344: so, in other words, shallow events constrained to the NwAFC, and deep events relatively ubiquitous, right? If yes, that might be a more direct, clearer phrasing.

As discussed in our response to comment 3, this section will be removed.

36. L. 347-351: The phrasing here is a little too vague here. It would be great to have some quantification, if possible.

As discussed in our response to comment 3, this section will be removed.

37. L. 350: "showing relatively higher OHC compared to years with shallow events in the BSO. Meanwhile, in the NwAFC region, except for 2011, the OHC is largely higher during shallow events compared to deep events." Is this robust? As mentioned in my main comments, on the figure, given your small event sample size, I don't see a clear separation between shallow and deep.

We agree that this result is not robust and requires further analysis, so we will remove this section. These results could be followed up in a future study.

38. L. 356-359: This sentence starts by introducing your own results, but ends with a citation. Please state more clearly what is a new finding, and what part has been discussed in other papers.

We will remove this reference to just focus on the results of the paper in the first paragraph of the discussion.

39. L. 361-362: "Our results also suggest that deep events received heat from the Norwegian Atlantic Slope Current (NwASC)," I am confused by this. Which part of the results showed this causal relationship?

As discussed in our response to comment 3, this section will be removed.

40. L. 365-380: Have you considered detrending SST instead to compute SSTA? It does not necessarily need to be computed, but given the strong Arctic trends, it immediately comes to mind when discussing baselines and MHW definition approaches (and could be worth discussing, in my view).

Thank you, this is an important point. We have discussed detrending SST as part of the MHW detection but chose to not detrend. The main reason for this is that detrending assumes that the increase in temperature in the Arctic is linear. In the Arctic Ocean and surrounding seas, the trend is not necessarily linear, as there is large natural variability. One example of natural variability in our region is decadal thermohaline anomalies (Årthun et al., 2017; Passos et al., 2024), which influence Atlantic inflows to the Nordic Seas on multi-year to decadal scales (Chafik et al., 2025). We will add a section on this to the discussion.

41. L. 382: observe -> find (as it is not observed here, or not directly)

Thank you, we will change this.

42. L. 384: "described in Mohamed et al. (2022); Barton et al. (2018)" I know the authors mean that the Polar Front is described in those papers, but it is not exactly clear here. Same as above, it would be very beneficial to discriminate what is your results vs. what is previous work. I would suggest moving these citations & the description of the polar front to the introduction.

We will revise the manuscript to clearly distinguish between our results and previous work. Thank you for the suggestion. We agree that the polar front is an important feature; however, the description of the polar front we provide is closely tied to the interpretation of the results in Section 3.1. For this reason, we feel it fits more naturally within the discussion rather than the introduction.

43. L. 388: to my understanding, "momentum mixing" is an atmospheric process. It could be worth explaining what it is and why it is relevant here.

As referred to in Raj et al. (2019), momentum mixing is when efficient turbulent convection transfers momentum down to the surface (Wallace et al., 1989). Momentum mixing destabilises air over warm water, and the increased turbulent mixing of momentum accelerates near-surface winds (Raj et al., 2019). We will revise the text to explain better why there is a high number of MHWs at the Mohn Ridge. We believe that the high number of MHWs at the Mohn Ridge could instead be attributed to high seasonal and interannual variability in the region, as detailed in Akhtyamova and Travkin (2023), shown to be highly correlated with the North Atlantic Oscillation (NAO). Another reason could be high temperature gradients associated with the interaction of warm and saline Atlantic waters with cold and fresher Arctic waters at the front (Akhtyamova and Travkin 2023). Lastly, lateral exchanges, such as eddies, across the front could trigger MHW events by bringing warmer water into a normally colder water region. We will address this in the text.

44. L. 391: There is some literature citation missing on the Atlantification in the Eurasian Basin, such as:

• Polyakov et al. (2017), Greater role for Atlantic inflows on sea-ice loss in the Eurasian Basin of the Arctic Ocean. 1126/science.aai8204

Thank you, we will refer to this paper by discussing that Atlantification is observed beyond Svalbard West.

45. L. 415-417: I like this justification. However, I think it could be useful to have it earlier, so the reader is not left wondering why a 10-day duration was picked in the first place.

Thank you, clarification for this choice will be added to the methods section.

46. L. 421-430: Malan et al. (2025) recently proposed a new, detailed classification of MHW vertical structure. If possible, it would be great to see this discussed here (See full reference provided at the beginning of this review).

Thank you, we will discuss this new classification of MHW vertical structure in the discussion and compare its methods with those used in our study. This section will be combined with discussion on 'invisible marine heatwaves', for example, bottom events that are not detected by our study.

47. L. 454-455: Still cannot find where this is established (and I do not see this clearly in Figure 10)

As discussed in our response to comment 3, this section will be removed.

**References:**

Akhtyamova, A. and Travkin, V.: Investigation of Frontal Zones in the Norwegian Sea, Physical Oceanography, 30, https://doi.org/10.29039/1573-160X-2023-1-62-77, 2023.

Årthun, M., Eldevik, T., Viste, E., Drange, H., Furevik, T., Johnson, H. L., and Keenlyside, N. S.: Skillful Prediction of Northern Climate Provided by the Ocean, Nature Communications, 8, 15 875, https://doi.org/10.1038/ncomms15875, 2017.

Chafik, L., Årthun, M., Langehaug, H. R., Nilsson, J., and Rossby, T.: The Nordic Seas Overturning Is Modulated by Northward-Propagating Thermohaline Anomalies, Communications Earth & Environment, 6, 573, https://doi.org/10.1038/s43247-025-02557-x, 2025.

Hobday, A. J., Alexander, L. V., Perkins, S. E., Smale, D. A., Straub, S. C., Oliver, E. C., Benthuysen, J. A., Burrows, M. T., Donat, M. G., Feng, M., Holbrook, N. J., Moore, P. J., Scannell, H. A., Gupta, A. S., and Wernberg, T.: A hierarchical approach to defining marine heatwaves, Progress in Oceanography, 141, 227–238, https://doi.org/10.1016/j.pocean.2015.12.014, 2016.

Lien, V. S., Raj, R. P., and Chatterjee, S.: Surface and bottom marine heatwave characteristics in the Barents Sea: a model study, State of the Planet, 8, https://doi.org/10.5194/sp-4-osr8-8-2024, 2024.

Malan, N., Gupta, A. S., Schaeffer, A., Zhang, S., Doblin, M. A., Pilo, G. S., Kiss, A. E., Everett, J. D., Behrens, E., Capotondi, A., Cravatte, S., Hobday, A. J., Holbrook, N. J., Kajtar, J. B., and Spillman, C. M.: Lifting the Lid on Marine Heatwaves, Progress in Oceanography, 239, 103 539, https://doi.org/10.1016/j.pocean.2025.103539, 2025

Mohamed, B., Nilsen, F., and Skogseth, R.: Marine Heatwaves Characteristics in the Barents Sea Based on High Resolution Satellite Data (1982–2020), Frontiers in Marine Science, 9, https://doi.org/10.3389/fmars.2022.821646, 2022.

Passos, L., Langehaug, H. R., Årthun, M., and Straneo, F.: On the Relation between Thermohaline Anomalies and Water Mass Transformation in the Eastern Subpolar North Atlantic, Journal of Climate, 37, 4821–4834, https://doi.org/10.1175/JCLI-D-23-0379.1, 2024.

Raj, R. P., Chatterjee, S., Bertino, L., Turiel, A., and Portabella, M.: The Arctic Front and its variability in the Norwegian Sea, Ocean Science, 15, 1729–1744, https://doi.org/10.5194/os-15-1729-2019, 2019.

Richaud, B., Hu, X., Darmaraki, S., Fennel, K., Lu, Y., and Oliver, E. C.: Drivers of Marine Heatwaves in the Arctic Ocean, Journal of Geophysical Research: Oceans, 129, https://doi.org/10.1029/2023JC020324, 2024.

Schlichtholz, P.: Relationships between Wintertime Sea Ice Cover in the Barents Sea and Ocean Temperature Anomalies in the Era of Satellite Observations, Journal of Climate, 34, 1565–1586, https://doi.org/10.1175/JCLI-D-20-0022.1, 2021.

Skogseth, R., Olivier, L. L., Nilsen, F., Falck, E., Fraser, N., Tverberg, V., Ledang, A. B., Vader, A., Jonassen, M. O., Søreide, J., Cottier, F., Berge, J., Ivanov, B. V., and Falk-Petersen, S.: Variability and decadal trends in the Isfjorden (Svalbard) ocean climate and circulation—An indicator for climate change in the European Arctic, Progress in Oceanography, 187, https://doi.org/10.1016/j.pocean.2020.102394, 2020

Smith, K. E., Sen Gupta, A., Amaya, D., Benthuysen, J. A., Burrows, M. T., Capotondi, A., Filbee-Dexter, K., Frölicher, T. L., Hobday, A. J., Holbrook, N. J., Malan, N., Moore, P. J., Oliver, E. C. J., Richaud, B., Salcedo-Castro, J., Smale, D. A., Thomsen, M., and Wernberg, T.: Baseline Matters: Challenges and Implications of Different Marine Heatwave Baselines, Progress in Oceanography, 231, 103 404,https://doi.org/10.1016/j.pocean.2024.103404, 2025.

Wallace, J. M., Mitchell, T. P., and Deser, C.: The Influence of Sea-Surface Temperature on Surface Wind in the Eastern Equatorial Pacific: Seasonal and Interannual Variability, Journal of Climate, 2, 1492 – 1499, https://doi.org/10.1175/15200442(1989)002<1492:TIOSST>2.0.CO;2, 1989.